# Revealing the chemical characteristics of Arctic low-level cloud residuals – in situ observations from a mountain site

Yvette Gramlich[1,2], Karolina Siegel[1,2,3], Sophie L. Haslett[1,2], Gabriel Freitas[1,2], Radovan Krejci[1,2], Paul Zieger[1,2], and Claudia Mohr[1,2,a]

[1]Department of Environmental Science, Stockholm University, Stockholm, Sweden
[2]Bolin Centre for Climate Research, Stockholm University, Stockholm, Sweden
[3]Department of Meteorology, Stockholm University, Stockholm, Sweden
[a]now at Department of Environmental System Science, ETH Zurich and Laboratory of Atmospheric Chemistry, Paul Scherrer Institute, Villigen, Switzerland

*Correspondence to*: Claudia Mohr (claudia.mohr@psi.chelaudia.mohr@aces.su.se)

**Abstract.** The role aerosol chemical composition plays in Arctic low-level cloud formation is still poorly understood. In this study we address this issue by combining in situ observations of the chemical characteristics of cloud residuals (dried liquid cloud droplets or ice crystals) and aerosol particles from the Zeppelin Observatory in Ny-Ålesund, Svalbard (approx. 480 m a. s. l.). These measurements were part of the one-year long Ny-Ålesund Aerosol and Cloud Experiment 2019-2020 (NASCENT). To obtain the chemical composition of cloud residuals at molecular level, we deployed a Filter Inlet for Gases and AEROsols coupled to a Chemical Ionization Mass Spectrometer (FIGAERO-CIMS) with iodide as the reagent ion behind a Ground-based Counterflow Virtual Impactor (GCVI). The station was enshrouded in clouds roughly 15 % of the time during NASCENT, out of which we analyzed 14 cloud events between December 2019 and December 2020. During the entire year, the composition of the cloud residuals shows contributions from oxygenated organic compounds, including organonitrates, and traces of the biomass burning tracer levoglucosan. In summer, methanesulfonic acid (MSA), an oxidation product of dimethyl sulfide (DMS), shows large contributions to the sampled mass, indicating marine natural sources of cloud condensation nuclei (CCN) and ice nucleating particles (INP) mass during the sunlit part of the year. In addition, we also find contributions of the inorganic acids nitric and sulfuric acid, with outstanding high absolute signals of sulfuric acid in one cloud residual sample in spring and one in late summer (May 21 and September 12, 2020), probably caused by high anthropogenic sulfur emissions near the Barents and Kara Sea. During one particular cloud event, on May 18, 2020, the air mass origin did not change from before to during and after the cloud. We therefore chose it as a case study to investigate cloud impact on aerosol physicochemical properties. We show that the overall chemical composition of the organic aerosol particles is similar before, during, and after the cloud, indicating that the particles have already undergone one or several cycles of cloud processing before being measured as residuals at Zeppelin, and/or that on the timescales of the observed cloud event, cloud processing of the organic fraction can be neglected. Meanwhile there are on average fewer particles, but relatively more in the accumulation mode after the cloud. Comparing the signal of sulfur-containing compounds of cloud residuals with aerosols during cloud-free conditions, we find that sulfuric acid has a higher relative contribution to the cloud residuals compared to aerosols during cloud-free conditions, but we did not observe an increase in particulate MSA due to the cloud. Overall, the chemical composition, especially of the organic fraction of the Arctic cloud residuals, reflects the overall composition of the general aerosol population well. Our results thus suggest that most aerosols can serve as seeds for low-level clouds in the Arctic.

## 1 Introduction

Aerosol particles interact with solar radiation either directly by light scattering and/or absorption, or indirectly by acting as cloud seeds (Haywood and Boucher, 2000). On an annual scale, Arctic clouds have an overall warming effect on the surface (Shupe and Intrieri, 2004). The formation of a cloud particle requires the availability of sufficient water vapor and updraft to create supersaturated conditions, and aerosol particles that provide a surface for the water vapor to condense onto. This subfraction of aerosol particles is termed cloud condensation nuclei (CCN) or ice nucleating particles (INP). The physical and chemical properties of aerosols play an important role for describing aerosol-cloud interactions (Köhler, 1936; Seinfeld and Pandis, 2006). Better knowledge on their composition, especially of the subfraction activating as cloud droplets, helps to better constrain their effect in the warming Arctic, in addition to an improved understanding of their sources.

While in most of the regions on Earth water vapor is the limiting factor, in the Arctic the availability of aerosol particles able to act as CCN or INP can be limited (Mauritsen et al., 2011). Due to this limit in aerosol number concentrations, aerosol particles in the Aitken mode (diameter < 80 nm), even as small as 20 nm in diameter, are able to serve as CCN in the Arctic (Korhonen et al., 2008; Leaitch et al., 2016; Bulatovic et al., 2021; Pöhlker et al., 2021; Karlsson et al., 2021, 2022; Siegel et al., 2022).

Overall, the Arctic aerosol particle number, size and composition follows a distinct annual cycle, where long-range atmospheric transport dominates the accumulation mode particles in winter and spring, and frequent new particle formation in the summer the Aitken mode abundance. Fall is the cleanest season with the lowest particle number and mass concentrations, with only few accumulation mode particles (Tunved et al., 2013). local sources increase in importance during the summer. Relatively speaking, the aerosol composition is mainly dominated by sea salt and long range transport from lower latitudes in the winter, while organics of biogenic origin and sulfate are becoming increasingly important in late spring and summer (Moschos et al., 2022a). One specific phenomenon in the annual aerosol cycle is called "Arctic Haze", occurring in late winter and spring, when enhanced mass concentrations of aerosol particles mainly composed of long-range transported sulfate, particulate organic matter, and heavy metals, but also black carbon, nitrate, ammonium and dust (Quinn et al., 2007) are observed. Among the natural sources, the ocean plays an important role. Aerosol particles emitted by sea spray can comprise sea salt and organic material from the sea surface microlayer (Cavalli et al., 2004; Kirpes et al., 2019). Sea salt is the largest contributor to particulate matter by mass across the Arctic. Its relative contribution to total particulate matter has been shown to be higher in the dark period compared to the bright season (Moschos et al., 2022a). In addition to open ocean, the origin of sea salt has been attributed to blowing snow, especially during the dark season at high wind speeds over sea ice, e.g. in the central Arctic (Huang and Jaeglé, 2017). Model simulations from a decade ago predicted the sea salt emissions to increase with less sea ice cover (Struthers et al., 2011). An increasing trend in sea spray emissions was found recently at the Arctic monitoring station Ny-Ålesund on Svalbard, (Heslin-Rees et al., 2020); however, this trend was linked to changes in the circulation pattern rather than decreasing sea ice cover.

Also, theThe organic fraction of Arctic aerosol is dominated by anthropogenic sources in winter, and natural emissions increase in importance in summer. The primary source region for wintertime anthropogenic aerosol is Eurasia (Moschos et al., 2022b).

In summer, the growth of aerosol particles has been associated with the presence of methanesulfonic acid (MSA), produced from the oxidation of dimethylsulfide (DMS) released by marine phytoplankton, andmarine trimethylamine ($N(CH_3)_3$) and other organic compounds (Willis et al., 2016; Beck et al., 2021). The highest particulate MSA concentrations in Ny-Ålesund were found to occur in May or June, when phytoplankton biomass is active in the surrounding Greenland and Barents Sea (Jang et al., 2021).

The presence of various particulate organic and sulfate-containing compounds was recently reported for the summertime Arctic Ocean using offline filter analysis (Siegel et al., 2021). These observations were among the first molecular-level measurements of semi-volatile aerosols from the high Arctic. Organic molecules with carbon chains of up to 18 carbon atoms were identified.

The largest signal was observed for compounds with 5 to 10 carbon atoms and 3 to 4 oxygen atoms. Semi-volatile organics were found to be involved in aerosol particle growth during summer in the marine Arctic environment (Burkart et al., 2017).

Although new information about the chemical composition of Arctic aerosol is emerging, very little is known yet about the composition of Arctic cloud residuals (dried liquid cloud droplets or ice crystals), from which information on the chemical composition of CCN and INP, aqueous-phase processing, or condensation of gaseous compounds (co-condensation, (Topping and McFiggans, 2012; Topping et al., 2013)) can be derived. Direct observations are scarce (Sect. S1, Fig. S1) and if available limited to intensive campaigns over just a couple of weeks; hence, there is no information on how the properties of cloud seeds

change throughout the year (McFarquhar et al., 2011; Wendisch et al., 2019). Aircraft observations in spring 2008 near Barrow (Alaska, ISDAC campaign) showed that compared to ambient aerosols, cloud residuals contained relatively less organics and more sea salt and black carbon (Hiranuma et al., 2013). However, no details on the organic molecular composition was reported. In addition, the cloud residuals were generally larger than the ambient aerosol particles (Zelenyuk et al., 2010; Hiranuma et al., 2013). In a case study during ISDAC, looking at the change in chemical composition of single particles at

different altitudes in a cloud, enriched sulfate was observed in the cloud residuals compared to the aerosol population below the cloud (Zelenyuk et al., 2010). Wendisch et al. (2019) measured cloud residuals by aircraft over Svalbard during the transition from spring to summer (end of May until beginning of June in 2017). They observed trimethylamine and sulfate in the cloud residuals, and to a lesser extent metals, organic carbon, and levoglucosan, a tracer for biomass burning. Higher levels of trimethylamine were found in cloud residuals sampled over sea ice, compared to over open ocean and drift ice. One very

recent study investigated the composition of cloud residuals in Ny-Ålesund based on 4 years of observations (Adachi et al., 2022). They focused on the fraction of sea salt, mineral dust, sulfate, K-bearing and carbonaceous material-containing particles and found that the cloud residuals have the same composition as ambient aerosol particles at positive temperatures, while at negative temperatures the residuals contained more mineral dust and sea salt compared to the ambient aerosol, likely reflecting the good INP ability of mineral dust and sea spray.

To the best of our knowledge, this study presents the first molecular-level observations of the chemical composition of Arctic cloud residuals. We identify organic and inorganic compounds in cloud residuals in the Arctic region using in situ measurements from a Chemical Ionization high-resolution time-of-flight Mass Spectrometer (CIMS) coupled to a Filter Inlet for Gases and AEROsols (FIGAERO), referred to as the FIGAERO-CIMS, set up behind a Ground-based Counterflow Virtual Impactor (GCVI) for a full year at the Zeppelin Observatory, Ny-Ålesund, Svalbard (Platt et al., 2022). We investigate the

changes in chemical composition of cloud residuals during different parts of the year, and of the aerosol population before, during and after one particular cloud event.

## 2 Methods

This section gives an overview of the instrumentation deployed during the one-year long Ny-Ålesund Aerosol and Cloud Experiment 2019-2020 (NASCENT) for this study, and the data processing procedures used to determine the chemical

characteristics of cloud residuals. Further information on the campaign and additional instrumentation used during NASCENT can be found in Pasquier et al. (2022).

### 2.1 NASCENT campaign

The data presented here were acquired during the one year long Ny Ålesund Aerosol and Cloud Experiment 2019 2020

(NASCENT). In brief, the aim of the campaign was to determine the physical and chemical properties of trace gases, aerosol and cloud particles in high detail with state-of-the-art instrumentation over the course of a full year. During the campaign,

measurements were conducted at different locations and altitudes near Ny-Ålesund, including the Zeppelin Observatory. In this study we only focus on the measurements conducted at the Zeppelin Observatory.

## 2.2 Zeppelin Observatory

The Zeppelin Observatory near Ny-Ålesund, Svalbard (78°54'N 11°53'E), is one of a few permanent measurement stations in the Arctic. The observatory itself is located about 2 km south of the small research settlement in Ny-Ålesund, on Mt. Zeppelin at an altitude of 474 m a.s.l., and is equipped with instrumentation to continuously measure atmospheric trace gases, particles and other atmospheric properties. The measurement station was established in 1989 and is now part of several monitoring programs as a global background station, e.g. the Global Atmosphere Watch Programme (GAW) (Platt et al., 2022). The

remote location of the observatory is characterized by the 26 km long Kongsfjorden located towards the north and east, and mountainous landscape with glaciers in the south and west. During the NASCENT year, low-level mixed phase clouds were present in all seasons, covering between 20 % to around 40 % of the monthly cloud cases (Pasquier et al., 2022). The Zeppelin Observatory itself is frequently covered in clouds (between October 2019 and December 2020 about 15 % of the time (visibility < 1000 m for at least 5 min)), providing an ideal location to study the physicochemical properties of Arctic clouds and aerosol-

cloud interactions.

## 2.3 Instrumental setup during NASCENT including GCVI

The instruments used in this study at the Zeppelin Observatory were connected through a three-way switching valve (Fig. 1) to a whole-air inlet during cloud-free conditions and to a GCVI (Brechtel Manufacturing Inc., USA, Model 1205) inlet during cloudy conditions. The inlet height was approx. 480 m.a.s.l.. With the whole-air inlet, both interstitial (non-activated) aerosol

particles and cloud droplets ~< 40 µm were collected, whereas the GCVI inlet only sampled cloud droplets and ice crystals > 6 to 7 µm (aerodynamic diameter). With our setup we cannot state how large the fractions of droplets and ice crystals were. The only available instrumentation to differentiate between a mixed-phase, liquid or ice cloud is the cloud radar. In brief, the working principle of the GCVI is as follows: within a wind tunnel, the cloud particles are accelerated onto the GCVI tip, where an opposing air flow (counterflow) is generated such that only those particles with high enough inertia make it into the sampling

flow, thereby removing the interstitial aerosol and particles < 6-7 µm (Ogren et al., 1985; Noone et al., 1988). This lower cut size of the GCVI was achieved by keeping the counterflow always one liter higher than the sampling flow, 16 and 15 L min⁻¹, respectively. Regular zero checks were done during operation where the counterflow was set close to 0 L min⁻¹ during cloud free conditions to ensure that the flow control operates well. The quality of the counterflow was also routinely tested by switching off the wind tunnel and only sampling only counterflow through the sample line. A more detailed description and

evaluation of the inlet system, the GCVI and additional instrumentation used to characterize microphysical aerosol and cloud residual properties is available in a previous study (Karlsson et al., 2021).

The number of cloud particles sampled with the GCVI is higher compared to the actual ambient cloud particle concentration. This enrichment of particles can be corrected for by calculating an enrichment factor (EF), which is determined by the settings of the GCVI (sampling flow and airspeed in the wind tunnel) and its geometry (Shingler et al., 2012). We used the median EF

(EF$_{med}$) of the respective sample to correct the DMPS1 data during times when we sampled cloud residuals. For the entire dataset discussed here, the EF$_{med}$ was in the range of 6.8 and 19.5 (mean 11.1, median 10.5, Table 1).

In addition, a sampling efficiency (eff) needs to be determined for particle number concentrations sampled behind the GCVI. eff depends on the cloud particle number size distribution and has been experimentally calculated for our site to be 0.46 (Karlsson et al., 2021).

During the NASCENT campaign, the GCVI was partly operated in manual mode and partly in automatic mode. Manual mode means that the GCVI was manually turned on in the presence of a cloud. As an indication of cloudy conditions, the a visibility sensor (Belfort, Model 6400) that comes with the GCVI was used. Similar to Karlsson et al. (2021), we used a visibility

threshold of 1000 m, where visibility below this threshold meant presence of a cloud according to the World Meteorological Organization's definition of fog (World Meteorological Organization, 2008). In automatic mode the GCVI was automatically

turned on whenever the visibility at the observatory was equal or less than 1000 m.

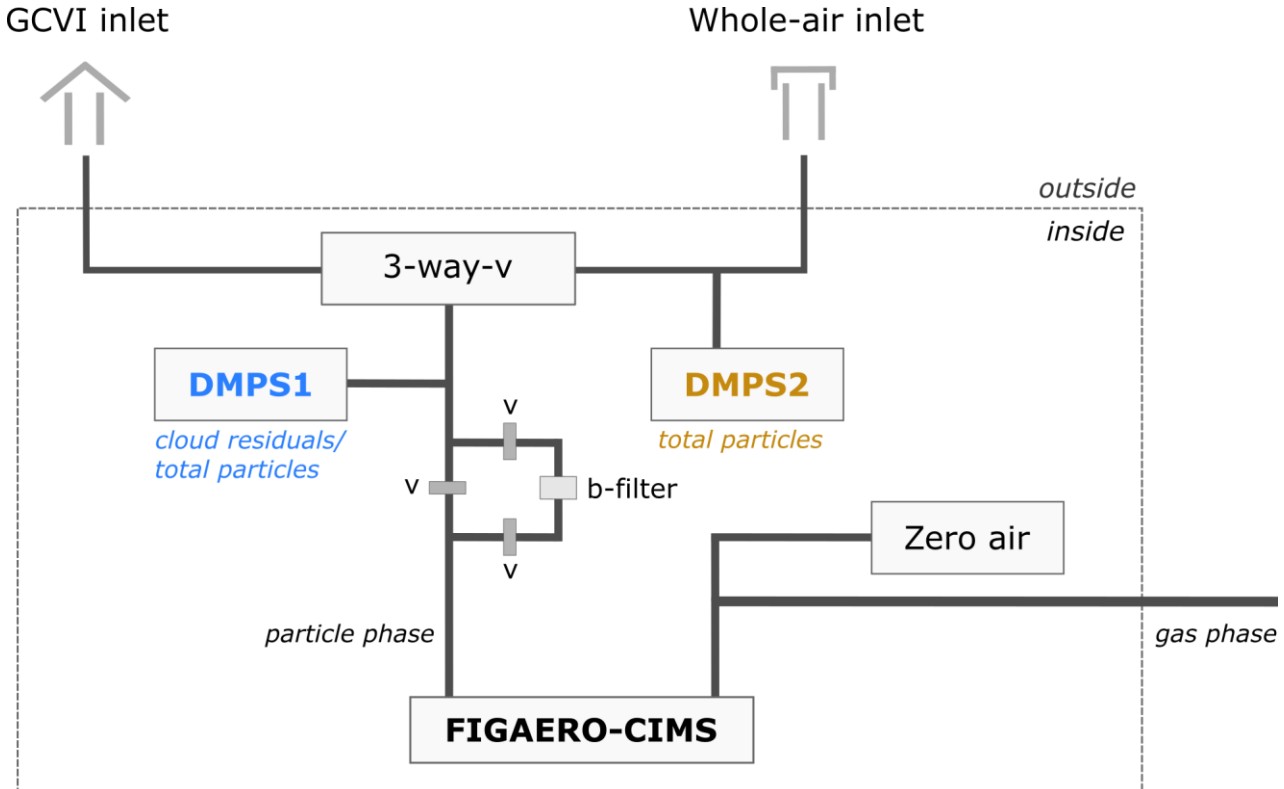

**Figure 1: Setup of the FIGAERO-CIMS and the DMPS at the Zeppelin Observatory. A three-way-valve (3-way-v) defines the type**
**of particles sampled. With the whole-air inlet, all particles (referred to as total particles, i.e. both aerosol particles and cloud particles) are sampled, whereas the GCVI inlet only samples cloud particles. Upstream of the FIGAERO-CIMS particle-phase inlet, a second filter (b-filter) was in place to perform regular particle-phase blanks. DMPS2 consists of two separate DMPS systems measuring partly overlapping size ranges. These two size ranges have been harmonized to one dataset in this study. Modified version of Fig. 1 from Karlsson et al. (2021), where DMPS2 consists of DMPS 2a and DMPS 2b.**

**2.4 FIGAERO-CIMS**

The chemical composition of cloud residuals at molecular level was obtained from a FIGAERO-CIMS (Aerodyne Research Inc., USA) (Lopez-Hilfiker et al., 2014; Thornton et al., 2020) using iodide (I⁻) as reagent ion. The FIGAERO-CIMS was installed in October 2019, and data is available almost continuously from November 2019 until December 2020, with a larger gap in July and August 2020.

The FIGAERO-CIMS was installed inside one of the rooms at the Zeppelin Observatory and connected to two separate inlet lines (Fig. 1). The particle-phase inlet was connected via ½ inch stainless steel tubing to the three-way valve of the main inlet switching between the whole-air inlet and the GCVI inlet. The gas-phase inlet was connected via ¼ inch PTFE tubing to ambient air directly through a hole in the wall.

The FIGAERO automatically cycled between its two modes: gas-phase analysis with simultaneous particle deposition, and
thermal desorption and analysis of the deposited particulate matter. These two modes are hereafter referred to as the sampling and heating periods, respectively. The sampling period with simultaneous particle deposition was 2.5 h. The particles were sampled with a flow of 4 LPM. During the gas-phase measurements, zero air was introduced every 40 min for 15 min, to obtain the gas-phase background signal.

After the sampling period, the particle-loaden filter was automatically moved to the desorption position. A 2 LPM flow of
heated ultrapure nitrogen (from a Nitrogen-generator, Peak Scientific NG5000) was passed through the filter to thermally

desorb the particles. This heating period was divided in three phases of 20 min each: ramping, soaking, and cooling. During ramping, the temperature of the nitrogen flow passing through the filter was gradually increased from room temperature to approx. 200°C. During the soaking phase, the heated nitrogen flow was held at this temperature to ensure complete evaporation of the deposited material. During the cooling phase, the nitrogen flow was cooled back down to room temperature. The signal obtained during the heating procedure (start of the ramping until end of the soaking) are hereafter referred to as thermograms. Particle-phase background samples, also called blanks, were collected using a filter upstream of the sampling filter (Fig. 1). Automatically switching valves allowed sample air to either bypass or pass through the zeroing filter (blank). Every third sampling period was a blank.

The FIGAERO-CIMS data was acquired at 1 s time resolution until February 2020 and at 2 s until the end of the campaign, and pre-averaged to 30 s for analysis. To account for the instrument´s sensitivity on the water vapor pressure in the ion-molecule reaction region, the data was normalized to the sum of iodide ($I^-$; mass-to-charge ratio (m/z): 126.905) and the iodide-water-cluster ($H_2OI^-$; m/z: 144.916) (Lee et al., 2014). Since dry nitrogen ($N_2$) is used to vaporize the particulate compounds during the heating period, accounting for the iodide-water-cluster signal is not relevant for the particle-phase data. Using iodide as reagent ion allows for mainly the detection of polar and oxygenated compounds (e.g. carboxylic acids), of which we can obtain the molecular composition up to an m/z of around 600 Th (Lee et al., 2014).

Since the cloud residual samples and blanks of this study were distributed over the entire year with long periods of cloud-free conditions and/or instrumental breaks in between, we treated the signals as if they were offline filter samples following the procedures detailed by (Cai et al., (2023). In order to take varying instrument backgrounds into account, for each compound, the signal of the blank thermogram was scaled by the signal ratio of blank and corresponding sample thermogram at the end of the soaking period (last 3-6 datapoints/last ~1.5 to 3 min, Method 2b in (Cai et al., 2023), Sect. S2, Fig. S2). After this step, the thermograms of the sample and the scaled blank were integrated from the beginning of the ramping period until the end of the soaking period. To obtain the total signal, the integrated thermogram signal from the scaled blank was subtracted from the integrated thermogram signal of the sample. The integration of the thermograms resulted in the total number of detected ions, which is also the unit of the data reported here.

To obtain the ambient aerosol composition (sampled via the whole-air inlet) of the two samples before and after the cloud for the case study of May 18, 2020, we used the neighboring ambient blank and subtracted the integrated thermogram blank from the integrated thermogram sample. We did not use the scaling approach here, as the instrument background of neighboring heatings usually does not differ substantially.

**Table 1: Cloud residual samples and the respective sampling start and end date and time (in UTC) of the FIGAERO-CIMS collection period, and corresponding median visibility, mean temperature (T) and median enrichment factor ($EF_{med}$) during the sampling time.**

| Cloud residual name | Sampling start | Sampling end | Median visibility (m) | Mean T (°C) | $EF_{med}$ |
|---|---|---|---|---|---|
| Dec 25, 2019 | 2019-12-25 16:42:29 | 2019-12-25 19:12:50 | 233.9 | -13.7 | 6.8 |
| May 18, 2020 | 2020-05-18 07:32:54 | 2020-05-18 10:03:02 | 48.8 | -3.1 | 9.7 |
| May 21, 2020 | 2020-05-21 12:47:44 | 2020-05-21 15:17:46 | 244.2 | -2.4 | 10.2 |
| Jun 2, 2020 | 2020-06-02 14:08:20 | 2020-06-02 16:38:22 | 135.3 | -0.5 | 10.6 |
| Jun 12-1, 2020 | 2020-06-12 18:20:20 | 2020-06-12 20:50:22 | 76.5 | -0.8 | 10.2 |

| | | | | | |
|---|---|---|---|---|---|
| Jun 12-2, 2020 | 2020-06-12 21:50:23 | 2020-06-13 00:20:25 | 265.3 | -1.9 | 10.6 |
| Jun 14, 2020 | 2020-06-14 01:50:04 | 2020-06-14 04:20:06 | 162.9 | -1.9 | 10.3 |
| Jun 25, 2020 | 2020-06-25 16:59:09 | 2020-06-25 19:29:11 | 87.7 | 4.2 | 9.6 |
| Jun 26, 2020 | 2020-06-26 17:29:11 | 2020-06-26 19:58:44 | 41.4 | -0.3 | 10.7 |
| Jun 27-1, 2020 | 2020-06-27 00:28:53 | 2020-06-27 02:58:55 | 101.1 | -0.5 | 12.0 |
| Jun 27-2, 2020 | 2020-06-27 03:58:56 | 2020-06-27 06:28:58 | 105.0 | -1.1 | 12.3 |
| Sep 12, 2020 | 2020-09-12 02:31:29 | 2020-09-12 05:01:39 | 168.9 | 1.2 | 9.3 |
| Oct 28, 2020 | 2020-10-28 20:38:37 | 2020-10-28 23:09:01 | 407.6 | -9.8 | 19.5 |
| Dec 9, 2020 | 2020-12-09 10:18:37 | 2020-12-09 12:49:04 | 432.6 | -4.2 | 14.1 |

## 2.5 DMPS

The particle number size distribution at the Zeppelin Observatory is continuously measured with a Differential Mobility Particle Sizer (DMPS) since 2000 (Ström et al., 2003; Tunved et al., 2013). In total there are three DMPS systems installed at the observatory, one, similar to the FIGAERO-CIMS, behind a three-way valve switching between the whole-air and GCVI inlets (DMPS1, installed in November 2015), and two behind the whole-air inlet. The two DMPS systems behind the whole-air inlet measure partly overlapping size ranges, which is why the data of these two were harmonized and are used hereafter

as one size range (DMPS2, Fig. 1). DMPS1 measured cloud residuals during cloudy conditions (when the GCVI was on) and ambient aerosol during cloud-free conditions (when the GCVI was off) in the size range of 10 to 945 nm. DMPS2 measured all ambient particles in the size range of 5 to 708 nm. More technical specifications of both DMPS systems can be found in Karlsson et al. (2021).

~~The number of cloud particles sampled with the GCVI is higher compared to the actual ambient cloud particle concentration.~~

~~This enrichment of particles can be corrected for by calculating an enrichment factor (EF), which is determined by the settings of the GCVI (sampling flow and airspeed in the wind tunnel) and its geometry (Shingler et al., 2012). We used the median EF (EF_med) of the respective sample to correct the DMPS1 data during times when we sampled cloud residuals. For the entire dataset discussed here, the EF_med was in the range of 6.8 and 19.5 (mean 11.1, median 10.5, Table 1).~~

~~In addition, a sampling efficiency (eff) needs to be determined for particle number concentrations sampled behind the GCVI.~~

~~eff depends on the cloud particle number size distribution and has been experimentally calculated for our site to be 0.46 (Karlsson et al., 2021).~~ All the cloud residual number size distributions in this study were corrected by a factor $k$ (Eq. 1), which is the inverse of the product of the sampling efficiency and the enrichment factor of the GCVI (see Sect. 2.3). No other particle loss calculations were applied. The data for the number size distributions of some samples was filtered for potential droplet splashing according to the procedure described in the supplementary (Sect. S3).

For the conversion from number to mass concentrations, we used a density of 1.3 g cm$^{-3}$, representing secondary organic aerosol (e.g. Alfarra et al., 2006; Malloy et al., 2009). The mass-based measurements (e.g. FIGAERO-CIMS) were not corrected by $k$, as we only show relative changes when comparing in cloud with out of cloud conditions, and no ambient concentrations when comparing the cloud residual samples. All given sizes in this study refer to the diameter (D).

$$k = \frac{1}{eff * EF_{med}}$$ (Eq, 1)

## 2.6 Back trajectories, meteorological data and cloud target classification

Air mass back trajectories were obtained using HYSPLIT (Stein et al., 2015), starting at 474 m a.s.l. at the Zeppelin Observatory (3-hourly archive data from the Global Data Assimilation System (GDAS), operated by the National Centers for Environmental Prediction (NCEP), 1 degree horizontal grid resolution). The back trajectories were calculated for 10 days, out of which the most recent 5 days were used here. Data within and above the boundary layer were used. The hourly temperature (T) and relative humidity (RH) data was downloaded from EBAS (https://ebas-data.nilu.no). The cloud target classification for Ny-Ålesund (Nomokonova et al., 2019) was taken from Cloudnet (https://cloudnet.fmi.fi/).

Table 1: Cloud residual samples and the respective sampling start and end date and time (in UTC) of the FIGAERO-CIMS collection period, and corresponding median visibility, mean temperature (T) and median enrichment factor (EF$_{med}$) during the sampling time.

| Cloud residual name | Sampling start | Sampling end | Median visibility (m) | Mean T (°C) | EF$_{med}$ |
|---|---|---|---|---|---|
| Dec 25, 2019 | 2019-12-25 16:42:29 | 2019-12-25 19:12:50 | 233.9 | -13.7 | 6.8 |
| May 18, 2020 | 2020-05-18 07:32:54 | 2020-05-18 10:03:02 | 48.8 | -3.1 | 9.7 |
| May 21, 2020 | 2020-05-21 12:47:44 | 2020-05-21 15:17:46 | 244.2 | -2.4 | 10.2 |
| Jun 2, 2020 | 2020-06-02 14:08:20 | 2020-06-02 16:38:22 | 135.3 | -0.5 | 10.6 |
| Jun 12-1, 2020 | 2020-06-12 18:20:20 | 2020-06-12 20:50:22 | 76.5 | -0.8 | 10.2 |
| Jun 12-2, 2020 | 2020-06-12 21:50:23 | 2020-06-13 00:20:25 | 265.3 | -1.9 | 10.6 |
| Jun 14, 2020 | 2020-06-14 01:50:04 | 2020-06-14 04:20:06 | 162.9 | -1.9 | 10.3 |
| Jun 25, 2020 | 2020-06-25 16:59:09 | 2020-06-25 19:29:11 | 87.7 | 4.2 | 9.6 |
| Jun 26, 2020 | 2020-06-26 17:29:11 | 2020-06-26 19:58:44 | 41.4 | -0.3 | 10.7 |

| Jun 27-1, 2020 | 2020-06-27 00:28:53 | 2020-06-27 02:58:55 | 101.1 | -0.5 | 12.0 |
| Jun 27-2, 2020 | 2020-06-27 03:58:56 | 2020-06-27 06:28:58 | 105.0 | -1.1 | 12.3 |
| Sep 12, 2020 | 2020-09-12 02:31:29 | 2020-09-12 05:01:39 | 168.9 | 1.2 | 9.3 |
| Oct 28, 2020 | 2020-10-28 20:38:37 | 2020-10-28 23:09:01 | 407.6 | -9.8 | 19.5 |
| Dec 9, 2020 | 2020-12-09 10:18:37 | 2020-12-09 12:49:04 | 432.6 | -4.2 | 14.1 |


## 2.7 Definition of cloud residuals and selection of samples

In analogy to the previous study by Karlsson et al. (2021), we here define cloud residuals operationally as the particles remaining after drying the cloud droplets and ice crystals collected with the GCVI. This is to clarify that with our instrumental

setup, we do not measure CCN or INP directly. Even though cloud residuals are often termed as CCN or INP in the literature, this is not entirely correct, as coalescence or chemical reactions may occur inside the cloud particles and thereby modify the original cloud nuclei (Twohy and Anderson, 2008).

As mentioned above (Sect. 2.3), the GCVI was not operated automatically throughout the entire duration of the campaign. As a consequence, there are times when the visibility at the observatory was above 1000 m, but the GCVI was still on.

Following from the above we selected the FIGAERO-CIMS cloud residual samples according to the following criteria:

- The inlet was set to GCVI during the entire 2.5 h sampling time of the FIGAERO-CIMS, and GCVI flows were fully operational for at least 95 % percent of the sampling time.
- The median visibility during the 2.5 h sampling time was below 1000 m.

Following these criteria, we are left with in total 14 cloud residual samples and 10 cloud residual blanks from the NASCENT

campaign. An overview of the cloud residual samples and the respective ambient conditions during the sampling times are listed in Table 1.

## 3 Results and Discussion

### 3.1 Seasonal distribution of cloud residuals characterized by FIGAERO-CIMS

Figure 2 shows the monthly number of cloud residual samples and related blanks measured with the FIGAERO-CIMS during

NASCENT, as well as the corresponding monthly total number of hours with visibility below 1000 m, ambient temperature and RH. The monthly average temperature during the year-long campaign was lowest in March (-19°C) and highest in July (7°C) (Fig. 2a). Based on the temperature, most of our cloud residual samples are probably originating from mixed phase clouds. Compared to the meteorological average (1994-2018), the winter months of NASCENT were around 6 K colder, and the summer slightly warmer (around 2 K, Pasquier et al., 2022). Monthly average RH was lowest in December 2019 and

January 2020 (67 %), and highest in June and July (91 %, Fig. 2b).

Overall, the cloud occurrence at the measurement station was higher from February to September than during the winter months, as observed previously (Chernokulsky et al., 2017; Dekhtyareva et al., 2018; Gierens et al., 2020). March and June were the two months with the most hours of cloudy conditions at the observatory (Fig. 2c). Unfortunately, during that period, the GCVI was run mostly in manual mode, and therefore not always following the visibility threshold criterion. The sampling

time with the GCVI inlet was highest in June, resulting in total eight cloud residual samples and six blanks. In May we collected two samples and two blanks, and in September only one cloud residual sample and one blank. During October 2020 and December 2019, 2020, we could only measure one cloud residual sample in each month. Although the haze season (March-May) showed cloudy conditions (19 % of time), we did not measure cloud residual samples in this period due to problems with the instrumentation (in only 15 % of the cloudy conditions the GCVI was on).

In the following we will focus first on the number size distribution of ambient and cloud particles during NASCENT before we discuss their chemical composition in more detail.

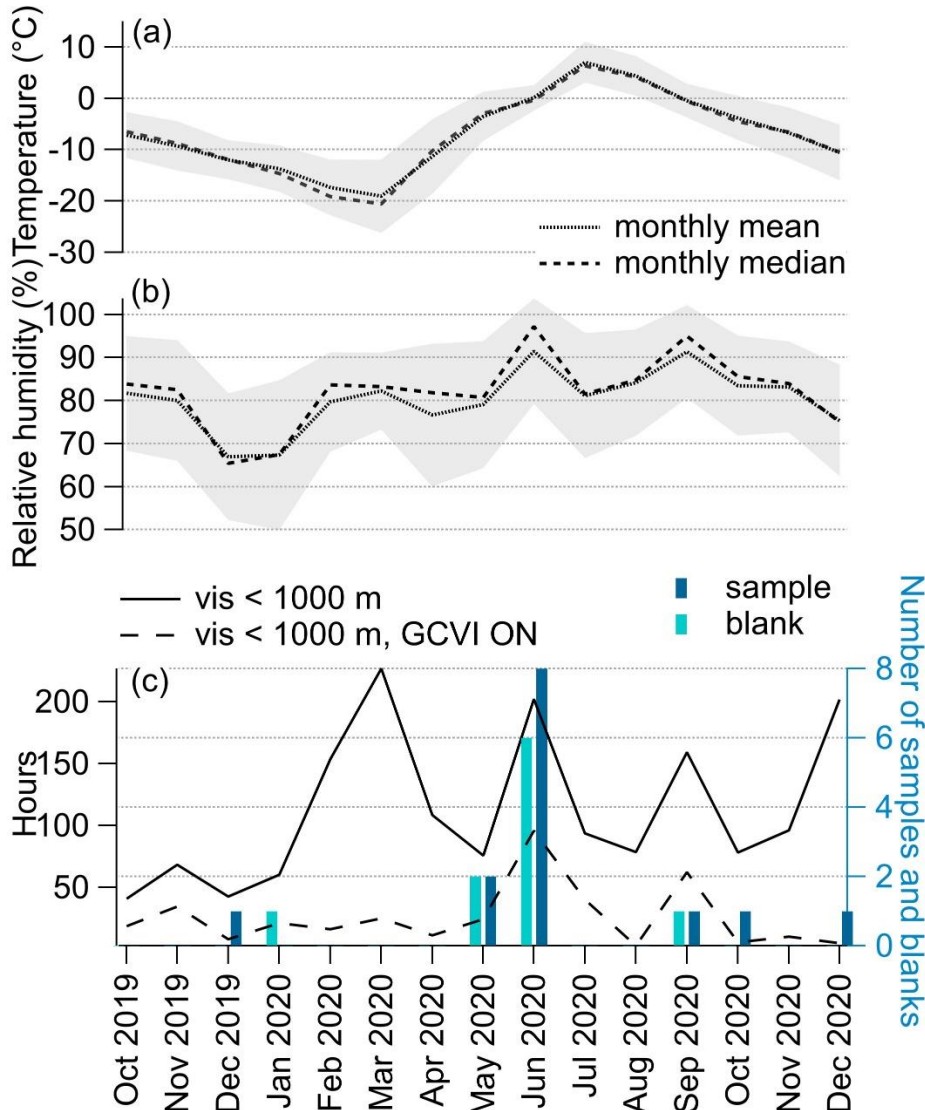

**Figure 2: Atmospheric conditions during the entire year of NASCENT measurement, and number of cloud residual samples. (a)**
**Monthly mean and median temperature, (b) monthly mean and median relative humidity. The grey shaded area represents the standard deviation. (c) Monthly total hours of visibility below 1000 m for at least 5 min (solid line), and corresponding total hours of sampling with the GCVI inlet (GCVI ON, dashed line), as well as the number of FIGAERO-CIMS cloud residual samples (sample) and blanks (blank) collected in the individual months.**

**3.2 Number size distributions of cloud residuals during NASCENT**

Karlsson et al. (2021) analyzed 2 years of cloud residual and corresponding total particle (cloud residuals and non-activated particles) sizes and number concentrations at the Zeppelin Observatory. They found that cloud residuals and aerosol particles

have a similar annual cycle with respect to number, with highest concentrations in spring and summer, and a minimum in fall

and winter. In the winter months (Dec-Feb) they observed a dominating fraction of Aitken mode particles (D < 80 nm) in the

cloud residuals, while during Arctic haze until summer (Apr-Jun) the cloud residuals were most frequently found in the

accumulation mode. Cloud residual sizes down to 20-30 nm were observed throughout all seasons. For liquid clouds, smaller

cloud residual sizes occurred at lower total particle numbers.

To illustrate how both the number and the size of the cloud residuals and the total particles behave for our cloud cases, we

show the number size distributions of cloud residuals and total particles for a cloud case of winter, spring, summer and late

fall (Fig. 3). In agreement with Karlsson et al. (2021), we observe more aerosol particles and cloud residuals in the spring and

summer, and fewer in winter and fall. Also, the ambient aerosol size distributions largely follow the findings from ~~their~~ a

previous long-term study on ambient aerosol number size distributions at this site (Tunved et al., 2013) with a ~~: the ambient~~

~~aerosol size distribution is~~ dominat~~ing~~ ~~ed by~~ accumulation mode in winter and fall, and a shift~~s~~ to ~~being~~ more Aitken mode

dominated distributions in summer. The number size distributions ~~Ff~~or the cloud residuals agree with ~~(~~Karlsson et al.~~,~~ (2021)

to some extent: they agree insofar that they both show accumulation mode cloud residuals dominating in spring~~, we observe~~

~~the largest sizes in spring~~ (144 nm on May 18, 2020), and Aitken mode cloud residuals dominating in winter (18 nm on Dec

25, 2020). In summer and late fall our cloud residuals show a peak in the Aitken mode (56 nm on Jun 12-1, 2020 and plateau

from around 66 to 144 nm in late fall), while ~~(~~Karlsson et al.,~~ ~~ (2021) show that cloud residuals in these seasons are more

dominated by accumulation mode particles. However, the cloud residuals in our study also exhibit a peak in the accumulation

mode in summer, with lower number concentrations compared to the Aitken mode. ~~and smaller sizes (56 nm on Jun 12-1,~~

~~2020) with a more pronounced Aitken mode in summer. In winter the cloud residuals show the smallest sizes (18 nm on Dec~~

~~25, 2020). The cloud residual sizes from late fall show a plateau in the average number size distribution from around 66 to 144~~

~~nm.~~

As we focus on a few cloudy events only throughout an entire year, deviations from the seasonal pattern of particle number

and size are expected. This can be illustrated in the case of the cloud event on June 27, 2020 (samples Jun~~e~~ 27-1 and Jun~~e~~ 27-

2, Sect. S3, Fig. S3), where the ambient average particle number concentrations (46 cm$^{-3}$) are much lower compared to typical

conditions during this time of the year (Ström et al., 2003; Tunved et al., 2013; Karlsson et al., 2021). The visibility was below

1000 m for several hours before we sampled the two cloud residual samples, and the low aerosol number concentrations are

most likely a result of cloud scavenging.

For most of the cloud samples, the average number of cloud residuals per size bin is smaller than the total number of particles

measured at the same time in the same size bin (Sect. S3, Fig. S3). This is largely expected, since the number of cloud residuals

is just a fraction of the number of total particles. However, for three cloud residual samples (Jun~~e~~ 27-1, Jun~~e~~ 25, and Sep~~tember~~

12, 2020), we observe on average more Aitken mode particles (14 nm - 16 nm) in the cloud residuals than the average total

particles. It is possible that this behavior is a sampling artifact from the GCVI. Karlsson et al., (2021) have discussed potential

artifacts in the cloud residuals due to shattering of ice crystals when hitting the tip of the GCVI, mainly during the winter

months when the fraction of ice crystals exceeds the number of liquid droplets. Our sample from December 25, 2019 (T: -

13.7°C) can be grouped in this category. In the Jun~~e~~ 27-1, Jun~~e~~ 25, and Sep~~tember~~ 12, 2020 samples, the temperature was

between -0.5°C and 4.2°C and the sampled clouds composed of mostly liquid droplets, but contained some ice as well (Sect.

S3, Fig. S4). In analogy to ice crystal shattering, it can be possible that the enhanced number of cloud residuals with a peak

below 20 nm arises from splashing drizzle droplets within the funnel of the GCVI wind tunnel. ~~Another~~ Other possible reasons

could be that in the dry counterflow air hygroscopic particles shrink to sizes much smaller than they have at ambient high

humidity conditions ~~shrinking of the cloud residuals in the GCVI when the water evaporates~~, the capture of smaller particles

by larger particles due to the wake effect (Pekour and Cziczo, 2011), or entrainment of drier air (Targino et al., 2007). Based

on the target classification on all the three days, it is likely that there was drizzle present (Sect. S3, Fig. S4).

These potential artifacts are of negligible relevance for the discussion of the chemical composition of the cloud residuals ~~the~~ following below, as their contribution to the bulk mass, and hence bulk chemical composition measured by the FIGAERO-CIMS is small (Sect. S3, Fig. S5). Moreover, in the case of droplet splashing, only the size of the cloud residuals is affected and not the chemical composition, as the material of the splashing droplets would remain in the cloud residuals when the cloud droplets evaporate.


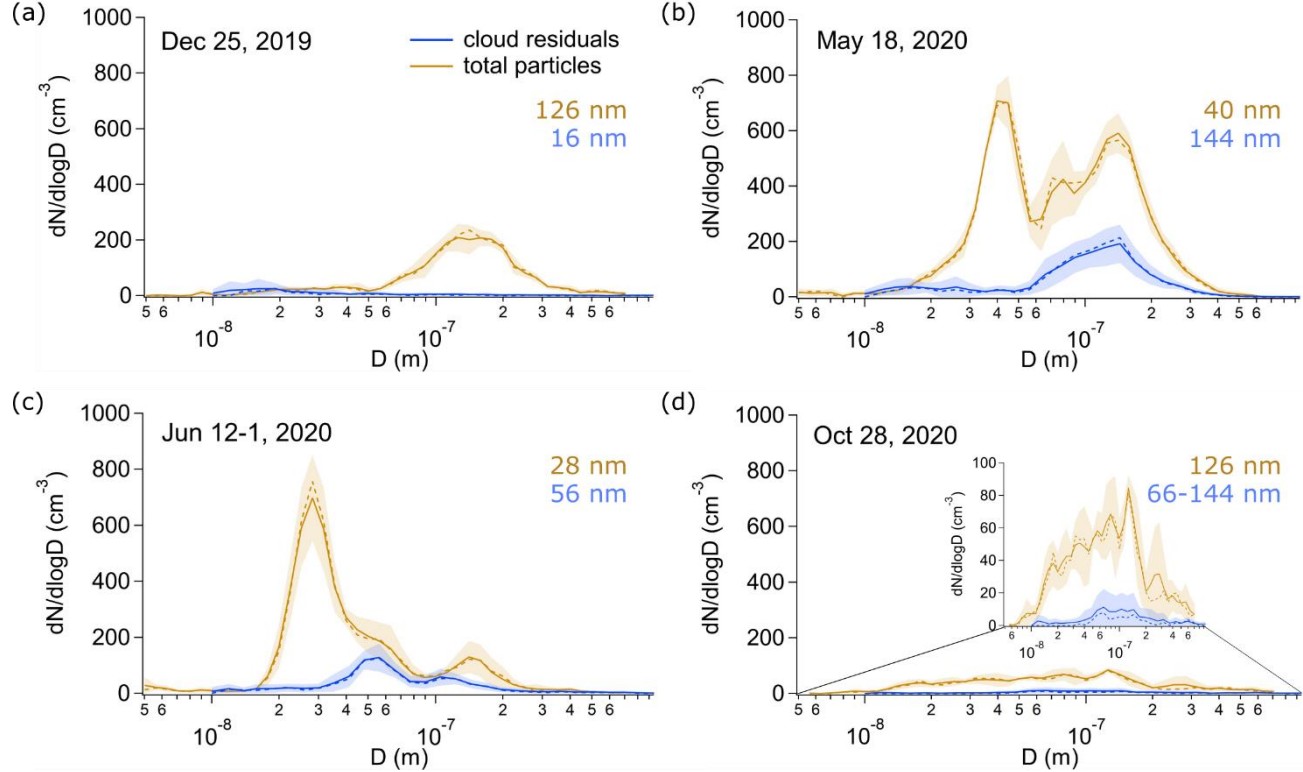

**Figure 3: Average number size distributions of cloud residuals and total particles during the corresponding 2.5 h FIGAERO-CIMS sampling time of (a) December 25, 2019 (winter), (b) May 18, 2020 (spring), (c) June 12-1, 2020 (summer), (d) October 28, 2020 (late fall). The insert in (d) shows the number size distributions of cloud residuals and total particles with the y-axis adjusted such that**
**the shapes of the distributions are visible. The solid and the dashed lines represent the mean and the median, respectively. The shaded areas are the standard deviations. The orange and blue numbers show the ~~mean~~ peak in the particle number size distribution of the total particles and the cloud residuals, respectively.**

### 3.~~3~~4 Chemical characteristics of cloud residuals

In the following we discuss the chemical composition of the cloud residuals of the 14 cloud events measured during the year-long NASCENT campaign with the FIGAERO-CIMS behind the GCVI as a function of season, and in relationship to the composition of the total particle population. With the FIGAERO-CIMS and using iodide as reagent ion, we were able to identify the molecular composition of in total 1558 different compounds. These include inorganic compounds, and organic compounds following the formula $I_hC_iH_jO_kY_l^-$ (Y = {N, S, Cl, Br, Si}), where $h$ is the number of iodide atoms, $i$ the number
of carbon atoms, $j$ the number of hydrogen atoms, $k$ the number of oxygen atoms, and $l$ the number of either nitrogen, sulfate, chloride, bromide or silicon atoms. In total 1094 of the detected compounds were clustered with iodide ($h$ = 1-3).

The observed compounds can be grouped as following based on their composition: ICHO ($h$ = 1, $i$ > 0, $j$ > 0, $k$ > 0), CHO ($h$ = 0, $i$ > 0, $j$ > 0, $k$ > 0), ICHON ($h$ = 1, $i$ > 0, $j$ > 0, $k$ > 0, Y = N, $l$ = 1), ICHOS ($h$ = 1, $i$ > 0, $j$ > 0, $k$ > 0, Y = S, $l$ = 1), I$_2$CHO ($h$ = 2, $i$ > 0, $j$ > 0, $k$ > 0), I$_3$CHO ($h$ = 3, $i$ > 0, $j$ > 0, $k$ > 0), and other. A subgroup of ICHO and CHO are likely fatty acids
(FA), with $h$ < 2, 3 < $i$ < 29, $k$ = 2, 0 < j < 2*$i$, representing a natural origin as they can be released into the atmosphere via sea spray emissions (Mashayekhy Rad et al., 2018). Levoglucosan (IC$_6$H$_{10}$O$_5^-$), a tracer for biomass burning, is part of the ICHO group. We exclude formic acid (ICH$_2$O$_2$) from the following analyses due to interference from the gas phase (Sect. S4, Fig.

S7). From the ICHON group we exclude $IC_6H_{15}O_7N$-. It is unclear where this compound is originating from, but since, as it has shows a very high signal in the cloud residual blanks, we attribute it to a background signal. For the ICHOS group, our focus hereafter lies on MSA (detected as $ICH_4O_3S$-, with a clear signal above background, Sect. S2, Fig. S2). In the "others" group we include the inorganic compounds sulfuric acid (SA, detected as $IH_2SO_4$-) and nitric acid (NA, detected as $IHNO_3$-), as well as organic compounds clustered with more than one iodide ion, $I_2CHO$ and $I_3CHO$, the identification of the molecular formula of which was not straightforward.

To illustrate the difference in the molecular composition of the cloud residuals observed during the different seasons of NASCENT we present a mass spectrum from winter (Dec 25, 2019), spring (May 18, 2020), summer (Jun 12-1, 2020) and late fall (Oct 28, 2020) each (Fig. 4). Negative signals occur due to the subtraction of the blank. For all seasons except winter, we find a clear contribution of organic compounds across the entire mass spectrum, and higher signals of a few individual compounds such as MSA, SA, and NA. For most of the cloud residuals in spring, summer and fall, the ICHO show a similar pattern to that in mass spectra from offline filter samples from the summertime high Arctic measured by FIGAERO-CIMS (Siegel et al., 2021), with several molecules in the mass range of about 220 until 360 Th. In contrast to spring, summer and fall, the winter cloud residuals show signal above background for only very few compounds, mainly NA and $IC_3H_6O_3$- (likely lactic aicd) and the fatty acids $IC_{16}H_{32}O_2$- (likely palmitic acid) and $IC_{18}H_{36}O_2$- (likely stearic acid). The two latter have been previously observed in the sea surface microlayer from the Arctic Ocean (Mashayekhy Rad et al., 2018). However, these observations were from the summer time high Arctic, and the conditions for winter might not be comparable. In addition, lactic, palmitic and stearic acid might also be attributed to handling of the GCVI during maintenance. We show the impact of these three compounds on the chemical composition of the cloud residuals in Sect. 3.4. In the dark months, the contribution of organic aerosol to the total particulate matter is expected to be lower than during the sunlit part of the year (Moschos et al., 2022b). The absence of more oxygenated organics in the wintertime cloud residual samples seems to reflect this. The average signal-weighted O:C ratio of the ICHO group is slightly larger than the ratio of the ICHON group (0.55 - 0.81, and 0.59 - 0.73, respectively), which is in agreement with offline FIGAERO-CIMS measurements conducted in the central Arctic (Siegel et al., 2021).

The cloud residuals observed in spring, summer and early fall (May 18, 2020 until Sep 12, 2020) show a clear MSA signal, whereas the cloud residuals observed in late fall and winter (Dec 25, 2019; Oct 28 and Dec 9, 2020) do not (Sect. S5, Fig. S8). The large signal of MSA suggests a marine contribution to CCN in the summertime. In a recent study from the Southern Ocean it was suggested that MSA is formed in the aqueous phase and thereby contributes to the growth of aerosol particles (Baccarini et al., 2021). In Sect. 2.73.5 we investigate this further.

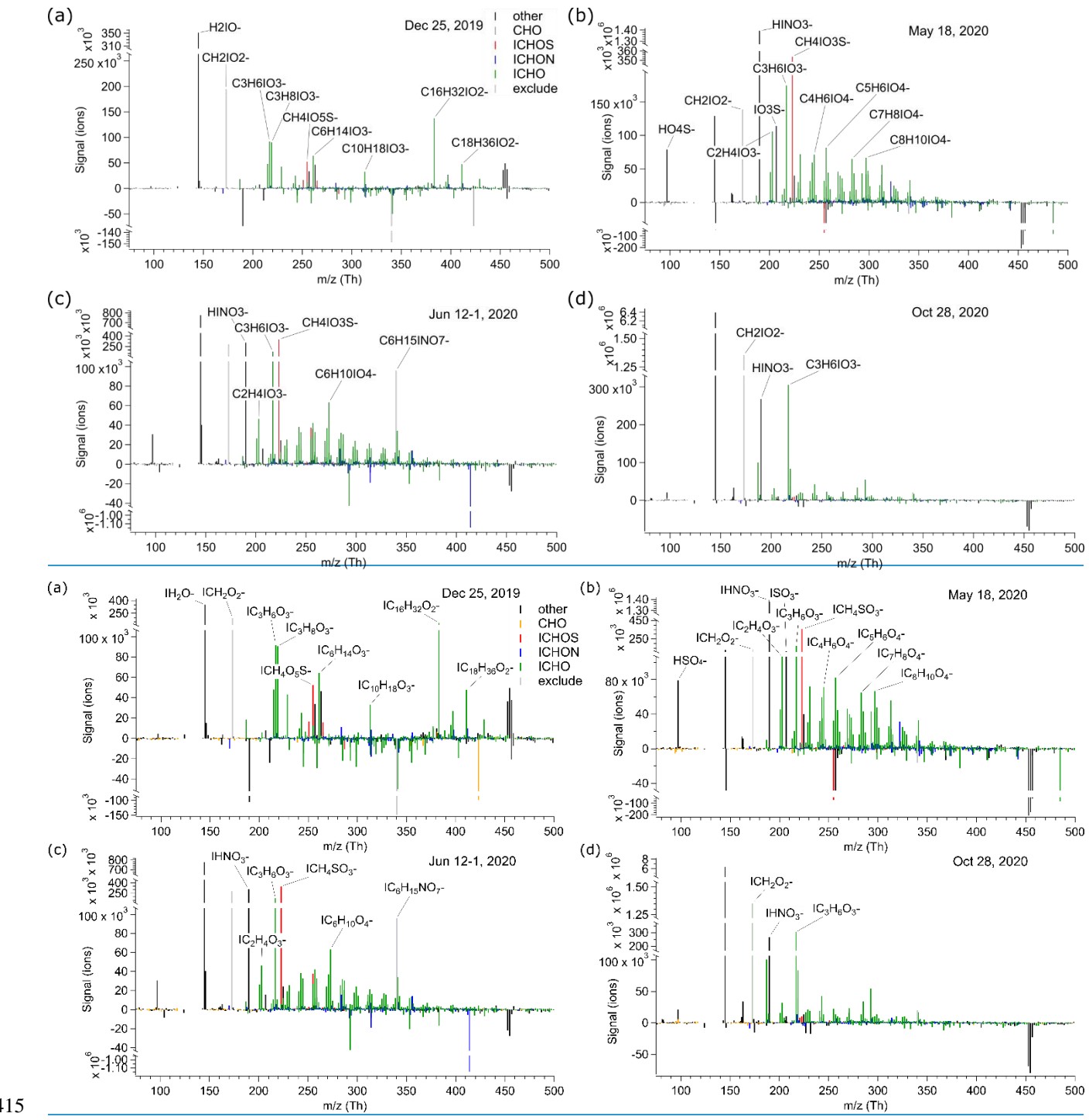

**Figure 4: Mass spectra of cloud residuals. (a) December 25, 2019, (b) May 18, 2020, (c) June 12-1, 2020, (d) October 28, 2020. The detected compounds are presented as I-clusters, as detected by the FIGAERO-CIMS.**

### 3.45 Seasonal patterns of cloud residual chemical components

In general, the ambient organic aerosol mass is much lower in the dark period of the year than during the bright season. Based on literature, Tthe particle composition in Ny-Ålesund during the dark period is dominated by anthropogenic emissions reaching the station due to atmospheric long-range transport, whereas in the sunlit period of the year, natural emissions can account for almost half of the organic submicron aerosol burden (Moschos et al., 2022a, b). The transition period from late winter to spring is known as the Arctic haze, characterized by enhanced aerosol mass concentrations (e.g. Tunved et al., 2013). In the following, we will investigate how this general pattern is reflected in the cloud residuals. We note that we have less cloud residual data during the winter months due to less cloudy conditions and lower concentrations close to the detection

limit, and in the haze month because of the mentioned problems with the instrumentation (Sect. 3.1); hence, our observations are skewed towards the summer months.

In Fig. 5 we present the cloud residual signal of various compounds and groups of compounds measured by FIGAERO-CIMS
behind the GCVI as absolute signal (Fig. 5a) and as relative signal (Fig. 5b). We include compounds related to natural (MSA, SA, levoglucosan, FA) and anthropogenic (ICHON, SA, NA) emissions and show that they are present in the cloud residuals throughout the year. We also indicate the absolute and relative signals for the winter samples when excluding the compounds that might be related to handling of the GCVI (lactic, palmitic and stearic acid). For more details about the signals of lactic, palmitic and stearic acid and that excluding these compounds does not change the overall pattern of the chemical composition
during the rest of the year see Sect. S6, Fig. S9, S10.

The largest fraction of compounds in all cloud residuals is made up by the organic CHO+ICHO groups. The relative contribution of these compounds to the total measured signal is highest in the two cloud residuals on Jun 12, whereas the absolute signal is highest in the cloud residual sample of Jun 25. Except for the sample of Sep 12, 2020, the observed absolute signal contribution follows the expected ambient organic aerosol seasonality (Moschos et al., 2022b). It is interesting to note
that from June 25 until September 12, the signal of CHO+ICHO and MSA both decrease in absolute numbers, but MSA has a more pronounced decrease in relative terms. This could be linked to the different source regions: While the MSA precursor DMS peaks in late spring until early summer (May-June), other organic compounds are present all year round and have both natural and anthropogenic continental sources (Behrenfeldt et al., 2008; Jang et al., 2021). The elevated levels of CHO+ICHO, NA and levoglucosan in the late summer (Sep 12) ~~in combination with the elevated levels of NA and levoglucosan at the same~~
~~time could~~ indicate an anthropogenic influence, supported by the back trajectories originating from the large anthropogenic source region of sulfur dioxide (SO$_2$) in Kola Peninsula (Sect. S7, Fig. S11).~~, or a combination of different sources the air masses encountered before arriving at the station.~~

The absolute signal of organonitrates (ICHON) follows the pattern of the CHO+ICHO group, and their relative contribution to the total signal observed with the FIGAERO-CIMS is similar for the individual cloud residual samples. ICHON
~~C~~compounds ~~in the ICHON group~~ might ~~have been formed via ICHO~~be related to oxidation~~s~~ of CH(O) and NOx emissions from year-round ship traffic in the surrounding seas and the Arctic Ocean, or from the nearby power plant (Eckhardt et al., 2013).

The seasonal contribution of compounds with molecular formulae corresponding to fatty acids with 4 to 28 carbon atoms follows the contribution of the CHO+ICHO. FA-like compounds were also found in ambient aerosol samples from the high
Arctic with offline FIGAERO-CIMS (Siegel et al., 2021). Their presence indicates organic enriched sea spray as a natural source of CCN (Mashayekhy Rad et al., 2018). The group of fatty acids also includes the previously mentioned palmitic and stearic acid that might be related to hygiene products. Excluding these two compounds would decrease both the absolute and the relative signal of fatty acids in Dec 25, 2020, but the pattern of the rest of the year remains similar (Sect. S6, Fig. S10).

The seasonal pattern of the absolute signal of NA in the cloud residual samples is similar to that of the CHO+ICHO group for
the majority of the samples. However, while CHO+ICHO has the highest absolute signal on June 25, NA shows the largest absolute and relative signal on May 18. Given that NA is related to anthropogenic emissions, and back trajectories (Sect. S7~~6~~, Fig. S11~~9~~) indicate the northern coast of Russia as the source region, it is possible that there was a prominent continental contribution to the aerosol population on May 18.

Levoglucosan (IC$_6$H$_{10}$O$_5^-$) is a known tracer for biomass burning (Simoneit et al., 1999). Of all the compounds and compound
groups we present in Fig. 5, levoglucosan has the lowest contribution to the observed signal in each of the cloud residual samples (below 1 % in all samples in Fig. 5b). We observe elevated absolute signals of levoglucosan in the cloud residuals ~~on May 21,~~ from mid-June until September 12 and lower absolute signals in May and October ~~also on December 9, 2020~~. These results show that long-range transport of biomass burning aerosol (Stohl et al., 2007; Zangrando et al., 2013; Moroni et al., 2020) is contributing to the CCN fraction during large parts of the year.

The measured total contribution of MSA to the cloud residual mass is overall low in the beginning of the year, increases towards June, and decreases again from September onwards. We observe the highest contributions to the cloud residual mass at the end of June 2020. Elevated MSA absolute signal can also be found in the cloud residuals in May, in the beginning of June and in mid-September 2020. The increase of MSA in the cloud residuals towards June follows the overall pattern of ambient particulate MSA levels measured with the FIGAERO-CIMS in 2020 (Siegel et al., 2023). Also in previous years, the
ambient concentration of particulate MSA in Ny-Ålesund was found to follow a seasonal pattern with the highest concentrations in May or June (Jang et al., 2021; Park et al., 2021; Moschos et al., 2022b). Since MSA is an oxidation product of DMS, the presence of MSA in the cloud residuals during the sunlit time of the year indicates a marine contribution to the aerosol particle population able to act as CCN. This ~~confirms~~ indicates that DMS oxidation products are relevant to grow aerosol particles to CCN-active sizes, which is what previous ambient aerosol observations at the same measurement location
~~suggesting~~ already suggested ~~that DMS oxidation products are relevant to grow aerosol particles to CCN-active sizes~~ (Park et al., 2021).

Another oxidation product of DMS is SA, which was observed in the cloud residuals as well. Since SA is formed via oxidation (in both gas and aqueous phase) of ~~sulfur dioxide (~~$SO_2$~~)~~, which has both anthropogenic (e.g. emissions from coal burning) and natural origins (e.g. oxidation product of DMS), source regions of the air mass need to be considered to estimate its origin.
The absolute signal of SA in general shows a similar pattern as the absolute signal of MSA, except for the cloud residuals on May 21 and Sep~~tember~~ 12, when both the absolute and relative SA signal is highest of all cloud residual samples. For May 21, HYSPLIT trajectories show that the air mass originated from the south-west and spent some time over the Greenland Sea, and the Barents and Kara Sea further east of Svalbard (Sect. S7~~6~~, Fig. S11~~9~~). On Sep 12, the trajectories show air arriving from the Barents Sea and the northern coast of Norway. The Greenland Sea south-west of Svalbard provides a source for DMS
emissions and air masses from this area have been previously observed to contain high levels of SA (Lee et al., 2020). A large anthropogenic source region of $SO_2$ emissions, the Kola Peninsula, is located near the coastal region of the Barents Sea. $SO_2$ emissions from this region have been identified to form SA, which drives new particle formation far away from the source region (Sipilä et al., 2021). Given that these high SA concentrations do not correlate with the observed MSA, the high SA signal in the cloud residuals on May 21 and Sep~~tember~~ 12 can probably be attributed to be mainly of anthropogenic origin. It
is interesting to note that on Jun 25, 2020, the HYSPLIT trajectories indicate air coming from near the continental border between Norway and Russia, similar to Sep~~tember~~ 12, but with less exposure over the Barents Sea. However, the cloud residual sample on Jun 25 contains much less SA and high MSA. The reason for this could be linked to the seasonality of MSA, with a large source in June (Jang et al., 2021). The different signals of the two DMS oxidation products suggests that the aerosol particles acting as CCN in the Arctic can have a strong seasonality, and natural sulfur sources might contribute to their mass
to a large extent during the phytoplankton bloom season.

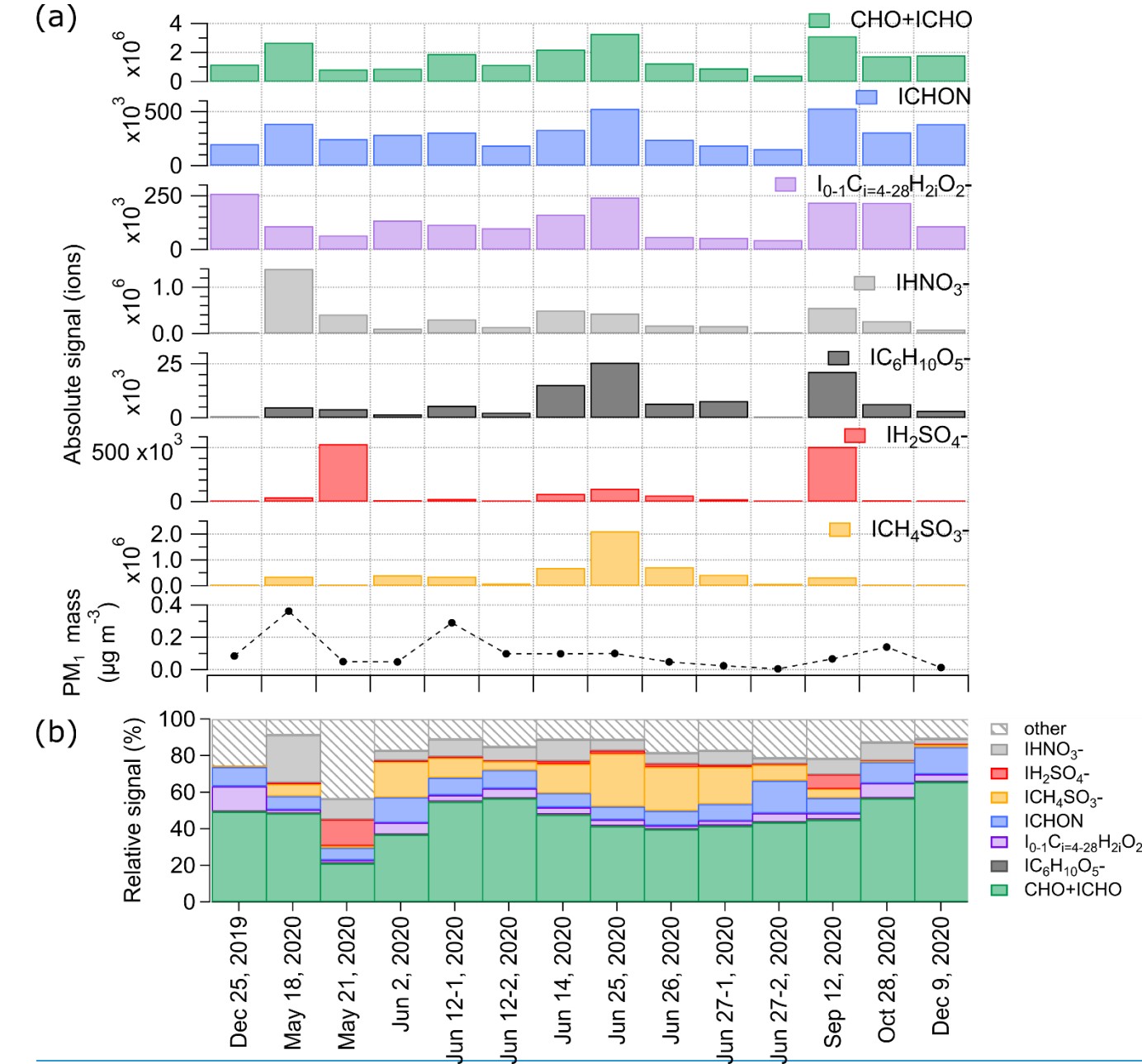

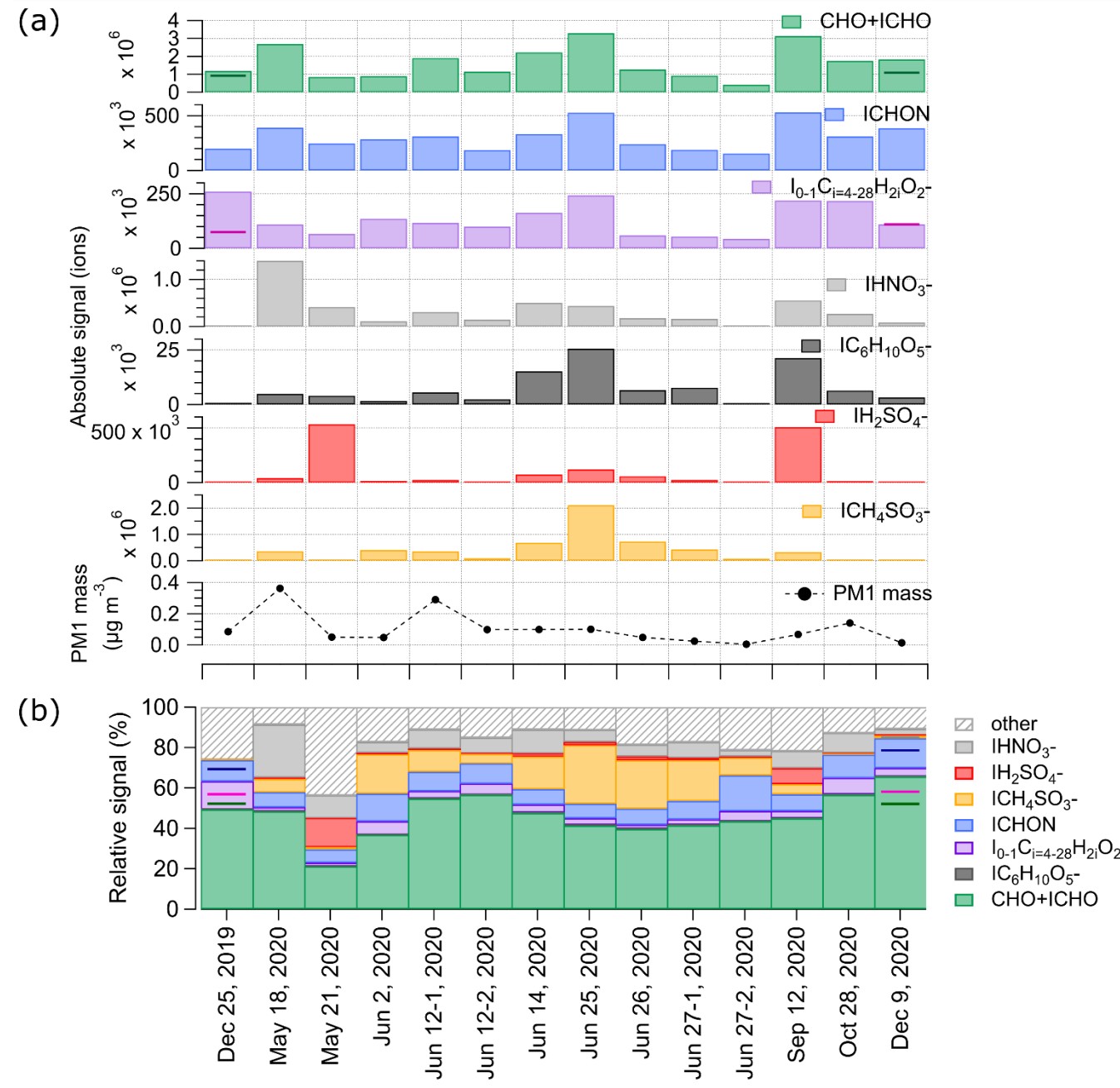

**Figure 5: (a)** Absolute signal of different compound groups (CHO+ICHO, ICHON, $I_{0-1}C_{i=4-28}H_{2i}O_2^-$ (fatty acids), $IHNO_3^-$ (NA $IC_6H_{10}O_5^-$ (levoglucosan), $IH_2SO_4^-$ (SA), $ICH_4SO_3^-$ (MSA)) in the different cloud residual samples, and the respective $PM_1$ mass. **(b)** Relative signal of different compound groups in the different cloud residual samples. Note: in the absolute signal view in (a) the CHO+ICHO group contains also the signal of $IC_6H_{10}O_5^-$, and $I_{0-1}C_{i=4-28}H_{2i}O_2^-$, whereas for the relative signal in (b) the signal from these two groups have been subtracted from CHO+ICHO. The colored horizontal lines in (a) indicate the absolute signals of CHO+ICHO and fatty acids when excluding the compounds that might be related to handling of the GCVI (lactic, palmitic and stearic acid). In analogy, in (b) the colored horizontal lines indicate the relative signal of CHO+ICHO (green), fatty acids (purple) and the ICHON (blue, see Sect. S6, Fig. S10 for the other compound groups) when excluding lactic, palmitic and stearic acid.

### 3.56 Cloud residual composition and size

To investigate if there is a link between the size and the sources of the cloud residuals, we investigated the mass fraction of Aitken mode particles (< 80 nm) to $PM_1$, and the number fraction of Aitken mode particles to the sum of Aitken and accumulation mode particles (Fig. 6a), and the ratio of MSA-to-SA (Fig. 6b). Since MSA is only produced from natural sulfur emissions and SA can be produced from both natural and anthropogenic emissions, their ratio provides a relative estimate of

the dominating source region. We note again that this is not a direct comparison of size and chemistry, as the overall contribution of the Aitken mode to the chemical composition is much smaller than that of the accumulation mode.

The number fraction of Aitken mode particles in the cloud residuals shows a minimum in spring and late fall (May 18 and Oct 28, 2020) and a maximum in the summer (Jun 27-1 and June 27-2, 2020) and in the winter (Dec 25, 20219 and Dec 9, 2020, Fig. 6a). The contribution of Aitken mode particles to the $PM_1$ mass of the cloud residuals is clearly increasing from spring towards summer (May 18 until Jun 27-2, 2020), and is low during the rest of the year. The dominating number of Aitken mode particles in the winter is most likely linked to the presence of ice particles that create artifacts in the GCVI (Karlsson et al.,

525 2021).

In general, the summer period with a high number contribution of Aitken-mode particles in the cloud residuals coincides with MSA/SA ratios > 10 and absolute MSA signals between around 75.000 and 200.000 ions in the cloud residuals high MSA signals (Fig. 6b). The highest MSA/SA ratio is found in the cloud residual sample on June 2, 2020, suggesting that during this cloud case the contribution of natural sulfate was highest of all the cloud residual samples. In absolute terms, the largest signal

of MSA is found in the cloud residual sample on Jun 25, 2020. Aitken-mode aerosol particles measured in Ny-Ålesund have previously been associated with new particle formation (NPF) events during the summer time when local sources dominate the aerosol population (Tunved et al., 2013; Lee et al., 2020). Ambient particulate MSA concentrations have been reported to be highest during NPF events, and based on gas-phase measurements, this DMS oxidation product has been shown to be an important contributor to the growth of newly formed particles in Ny-Ålesund (Dall´Osto et al., 2017; Beck et al., 2021; Park

et al., 2021). Aitken-mode particles have previously been shown to play a role as CCN in the summertime Arctic (Kecorius et al., 2019; Bulatovic et al., 2021; Karlsson et al., 2022). The combination of the observed dominating Aitken mode and elevated levels of MSA in the cloud residual samples at the end of June 2020 provides further indicates evidence that during this time of the year, MSA clearly contributes to the growth of newly formed particles into CCN sizes.


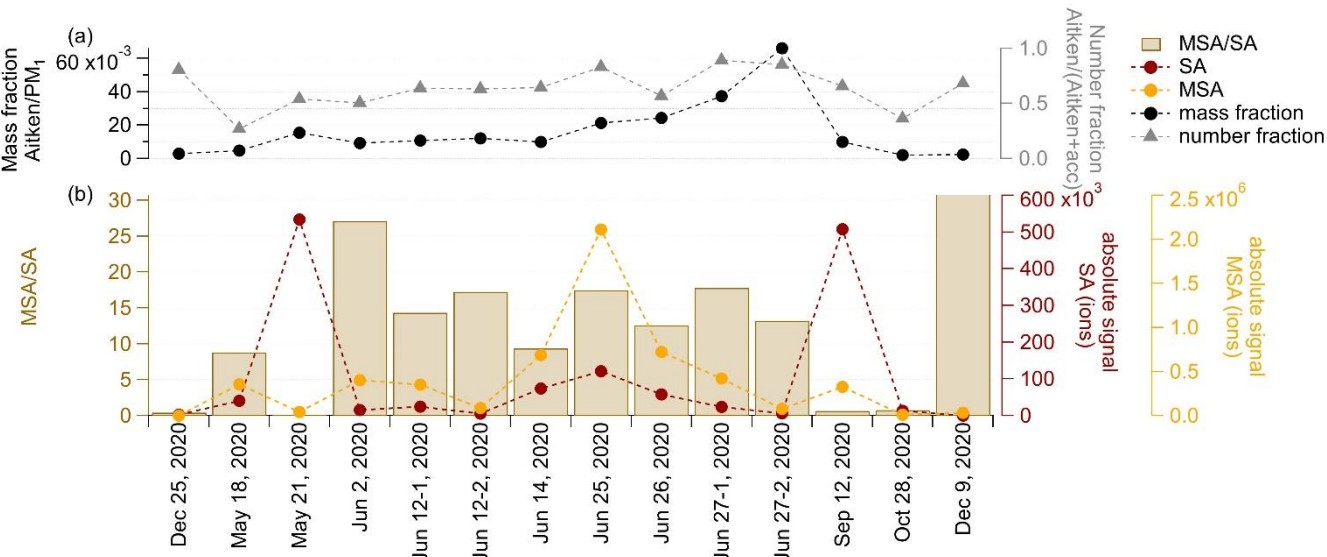

**Figure 6: (a) Mass fraction of Aitken mode particles to total $PM_1$, and number fraction of Aitken mode particles to the total number of Aitken and accumulation (acc) mode particles. As Aitken mode particles we consider particles < 80 nm. (b) Ratio of MSA to SA and their absolute signal for all cloud residuals.**


### 3.67 Cloud case study – May 18, 2020

In Fig. 7 we present different ambient properties during a cloud event on May 18, 2020. On this day we observed a cloud event before, during, and after which the air masses were originating from the same region (Fig. 7a). This situation allowed us to

study potential cloud processing of aerosol particles in detail. To do so, we identified FIGAERO-CIMS samples before, during, and after the cloud occurrence based on the observed visibility (Fig. 7b). *Before* defines the 2.5 h directly before the cloud was observed at the Zeppelin Observatory and the visibility was around 60 km. *During* is equal to the sampling time of the FIGAERO-CIMS cloud residual sample of May 18, 2020, measured in the beginning of the cloud when the median visibility was 49 m. For this time period we measured the size distribution of both the cloud residuals and the interstitial aerosol particles (together termed total particles), and the chemical composition of the cloud residuals only. *After* refers to the sampling time after the cloud had dissipated and the visibility was around 75 km. The average temperature during the cloud event was between -1.7 and -3.5°C, indicating most likely a low-level mixed-phase cloud, which is also identified as such in the target classification (Fig. 7c). The air mass reached the Zeppelin Observatory from the east, where it spent time over the northern coast of Russia. Throughout the cloud event the back trajectories show very similar source regions, with a slight shift to more marine influence over time. The air mass spent most of the time above the boundary layer (Sect. S87, Fig. S120). The accumulated precipitation along the back trajectories was on average 0.3 mm (median 0.2 mm, max. 1 mm, min. 0 mm).

### 3.67.1 Changes in aerosol number size distribution

*Before,* the aerosol number size distribution was composed of a dominating Aitken mode with a peak at around 45 nm and a smaller contribution of accumulation mode particles with a peak at around 125 nm (Fig. 7d). The total aerosol population also included particles in the nucleation mode (8-17 nm).

*During,* the accumulation mode particles with a peak at around 144 nm dominated the number concentration in the cloud residuals (Fig. 7d). A few Aitken mode particles down to around 56 nm were observed in the cloud residuals as well. Overall, the cloud residuals span a size range of around 60 to 300 nm. Due to the larger size of the accumulation mode particles, they are expected to activate into cloud droplets first. The peak in number occurring at 144 nm for the cloud residuals and lower numbers towards the 56 nm sized cloud residuals confirm this expected behavior. The total particle population (activated and non-activated aerosol particles) exhibited fewer particles in the nucleation and Aitken mode, and slightly more particles in the accumulation mode compared to *before*. Additionally, another peak in the size distribution occurred at around 79 nm in the total particle population, and a minimum at around 56 nm. The reduced number of nucleation and Aitken mode particles in the total particle population was probably a result of their coagulation with the cloud droplets. The second peak in the Aitken mode could indicate the advection of a different air mass. The back trajectories indicate that *during,* the air spent less time over the Russian coast compared to before, and more time over the Kara Sea (Fig. 7a). Additionally, while most of the time the air mass was above the boundary layer *before, during* and *after,* there are indications in the back trajectories that *during* the air was within the boundary layer when it was passing over the Kara Sea (Sec. S8, Fig. S12a, b). In this region, the air mass could have collected the particles visible as a peak in the number size distribution around 79 nm.

*After*, the number size distribution showed a bimodal distribution of Aitken and accumulation mode particles (Fig. 7d). Compared to *during,* the nucleation mode particles were completely removed and the number of Aitken mode particles had decreased further. Additionally, compared to *before*, the peak in the accumulation mode had shifted to a larger size at around 141 nm, similar to *during*. The average particle number concentration had decreased by about half, from 559 cm$^{-3}$ *before* to 344 cm$^{-3}$ *after*. *During*, there were on average 456 cm$^{-3}$ total particles and 105 cm$^{-3}$ cloud residuals. This decrease in total number can be mainly attributed to the decrease in Aitken mode particles. In combination with the shift to larger particles *after*, this could indicate growth of the accumulation mode particles due to coalescence with the Aitken mode particles *during*, which resulted in larger aerosol particles *after* when the cloud droplets had evaporated. It is also possible that the accumulation mode particles grew due to in-cloud processing. We investigate this in the next section.

The *after* bimodal size distribution with a minimum between Aitken and accumulation mode at 63 nm (known as the "Hoppel minimum" (Hoppel et al., 1986)) indicates a non-precipitating cloud, and agrees well with observations from the central Arctic

(64 to 71 nm, Karlsson et al., 2022). According to Hoppel et al. (1986), this minimum is related to the activation diameter of the aerosol particles. The presence of a Hoppel minimum in the aerosol size distribution at the Zeppelin Observatory has frequently been observed (Tunved et al., 2013).

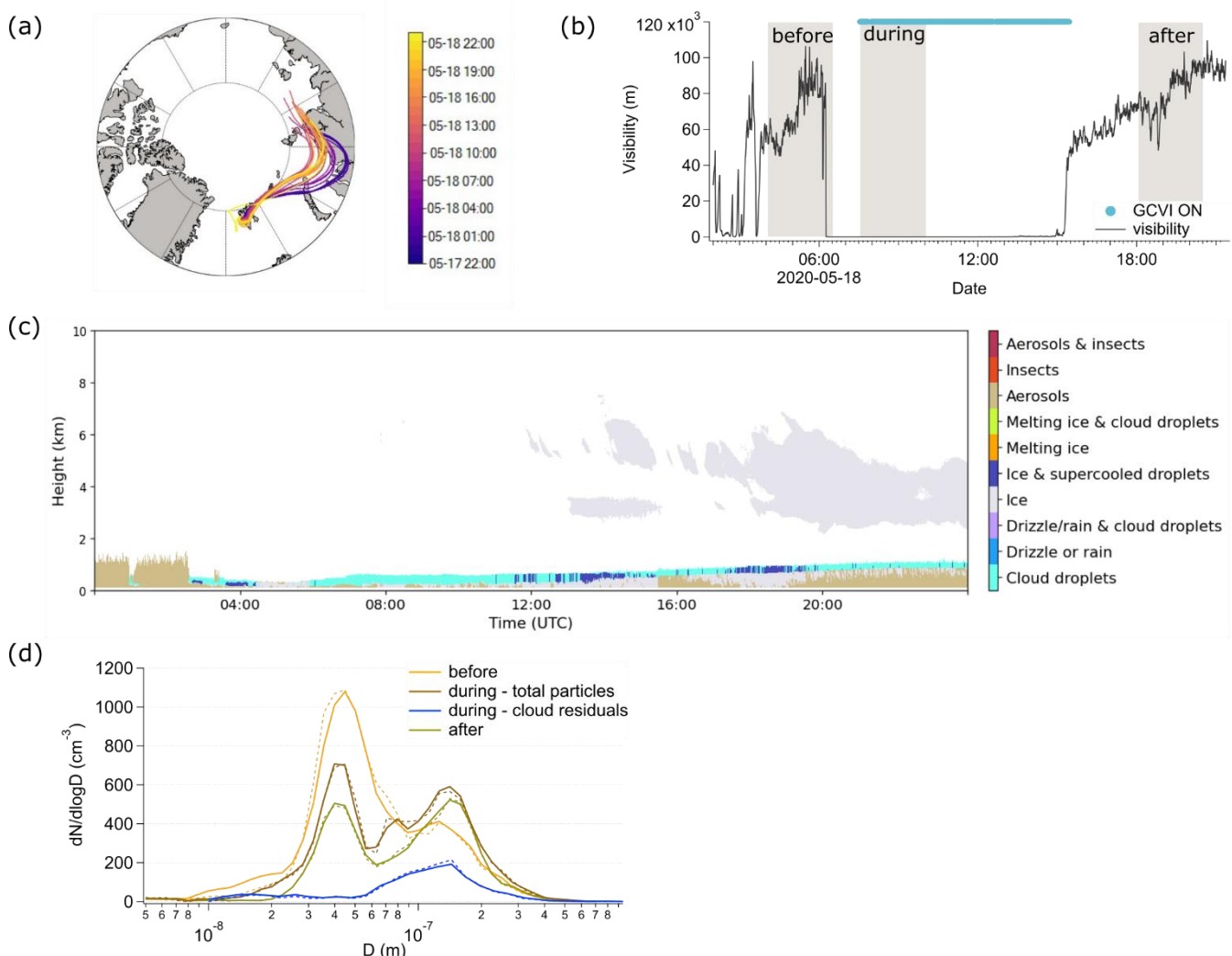


**Figure 7: Cloud case on May 18, 2020. (a) 5 days back trajectories of air masses arriving at the Zeppelin Observatory before, during and after the cloud event color-coded by time. (b) Time evolution of the visibility and sampling times of the FIGAERO-CIMS samples before, during and after the cloud event (grey shaded area). (c) Target classification on May 18, 2020 ((Cloudnet (2021), last access: 2022-11-20, 20:17 UTC) (d) Number size distributions before, during and after the cloud event. Before and after the cloud event the**
**two DMPS both measure behind the whole-air inlet; hence, they both measure the same particle population during these times. For before and after we only show the DMPS2 data. During the cloud event, DMPS1 measured behind the GCVI and DMPS2 measured behind the whole-air inlet; hence, DMPS1 measured the cloud residuals and DMPS2 the total particle population. Average total particle number concentrations: 552 cm³ before, 379 cm³ during – total particles, 96 cm³ during – cloud residuals, 285 cm³ after.**

**3.6̶7̶.2 Changes in aerosol chemical composition**

Overall, we observed only small changes in the chemical composition throughout the cloud event (Fig. 8a, pie charts). The "other" group, containing largely inorganic sulfur- and chloride-containing species, contributes the same fraction throughout the cloud event. The largest change from *before* to *after* is the increased fraction of CHO+ICHO (also in absolute terms, not shown here), whereas MSA and NA show a slight decrease. Also, the absolute signal of NA is decreasing from *before* to *after*
(not shown here). MSA and NA show a decreasing trend also from *before* to *during*. However, the changes in chemical composition are not that pronounced, which is similar to a study conducted at a mountain site in Sweden using a Quadrupole

Aerosol Mass Spectrometer behind a GCVI observing no significant change in aerosol composition between before and after a cloud event (Drewnick et al., 2006).

As the CHO+ICHO contribute the largest fraction of cloud residual signal for the cloud case on May 18, 2020, and also of most of the cloud residuals measured with the FIGAERO-CIMS during the year (see Sect. 3.45), we investigate this group in more detail and look at their molecular composition. Overall, the molecular composition of the CHO+ICHO does not change much during to the cloud event. The majority of the CHO+ICHO are composed of 2 to 11 carbon atoms and 3 to 6 oxygen atoms (Fig. 8a). Among these, the compounds with 4 oxygen atoms, likely dicarboxylic acids, dominate the signal *before*, *during* and *after*. Dicarboxylic acids have been shown to be part of the water-soluble fraction of organic aerosols (Saxena and Hildemann, 1996). *During,* the number of oxygen atoms shifts to smaller numbers. Most of the CHO+ICHO signal comes from compounds with 5 and 7 carbon atoms. The largest change is observed for compounds with 3 carbon atoms. Their relative contribution is much lower *before* than *during* and *after*. Compounds that show a smaller change in relative contribution are those with 4 and 10 carbon atoms. Their contribution is much higher *before* and *after* than *during* (Sect. S87, Fig. S131). The increase in absolute and relative signal *after* compared to *before*, is possibly linked to aqueous phase oxidation in the cloud droplets.

In the following we identify individual compounds that showed an increasing (79 compounds, hereafter termed as *increasing*) or decreasing (46 compounds, hereafter termed as *decreasing*) trend from *before* to *after* the cloud event, and compounds that are highest *during* in the cloud residuals (Fig. 8b, including all CHO+CHOI, CHION, NA, MSA, SA groups). In Fig. 8b we show the trends in form of a mass defect, to show how the compounds are spread over the entire m/z range. The *highest during* category comprises 428 compounds. Of all the compounds with a change in relative signal, no specific chemical families could be identified. We therefore focus on the three compounds showing the most pronounced relative change in signal in each of the three groups (*increasing, decreasing, highest during*).

The three compounds with the strongest relative increase were $IC_4H_6O_4-$, $IC_3H_4O_4-$, $IC_5H_8O_4-$, which are the chemical formulae corresponding toof succinic acid, malonic acid and glutaric acid, respectively. Succinic and malonic acid have previously been found in cloud water in continental Europe (Löflund et al., 2002). We also observed levoglucosan in this group. Except for succinic acid, these acids have been found to be able to serve as CCN in the laboratory (Cruz and Pandis, 1997; Giebl et al., 2002; Rosenørn et al., 2006). It is likely that the compounds in the *increasing* category are chemical tracers of good CCN, or produced in the aqueous phase through oxidation of water-soluble or co-condensing gases, and due to their higher oxidation state are more likely to stay in the particulate phase after the cloud droplets evaporate.

The three compounds that showed the strongest relative decrease were NA, $IC_2H_4O_3-$ (might be glycolic acid), and $IC_5H_6O_4-$ (might be citraconic acid or glutaconic acid). Their decrease could be due to either aqueous-phase reactions or evaporation.

The three compounds that showed the highest signal *during* were $IC_3H_6O_3-$, $IC_7H_8O_4-$ and $IC_8H_{10}O_4-$. $IC_3H_6O_3-$ can likely be attributed to lactic acid, whereas the other two compounds could not be unambiguously identified. The compounds in the *highest during* category may be particularly efficient as CCN or INP (ice nucleating particle), or be formed in the aqueous phase. However, lactic acid is also a tracer of human activity, and the high lactic acid signal could therefore indicate contamination from handling of the GCVI during maintenance.

The normalized signal of SA showed that it belongs to the category *highest during*. The elevated level of SA in the cloud residuals indicates aqueous-phase reaction of dissolved $SO_2$ in the cloud droplets (Hoppel et al., 1994). Elevated contributions of sulfate in the cloud residuals compared to the ambient aerosol particles were also found in an aircraft study during the haze season near Alaska (Zelenyuk et al., 2010). In contrast to our study, they measured the difference in ambient particle and cloud residual composition vertically, where ambient particles were measured below the cloud. It is interesting that the SA levels *after* drop down to a similar level as *before* (Fig. 8c). If SA is produced inside the cloud droplets, we would expect the evaporated cloud droplets *after* to be enriched with SA. SA is rather non-volatile and partitioning into the gas phase therefore unlikely. From the changes in the total particle number size distribution we saw an additional Aitken mode at around 79 nm

655   *during*, which we attributed to be linked to advection of a different air mass. Hence, a possible explanation for the lack of enrichment in SA after the cloud could be that we were not measuring the same air mass throughout the cloud event, or different air mass got entrained from above throughout the cloud event.

MSA does not show a clear trend from *before* towards *after* (Fig. 8b). It showed the highest normalized signal *before* and the lowest *during* (Fig. 8c). Hence, our data does not support previous studies stating that MSA is mainly produced in the aqueous

660   phase (Hoffmann et al., 2016; Chen et al., 2018; Baccarini et al., 2021; Xavier et al., 2022). The ratio of MSA to SA *before* and *after* is very similar, whereas *during* the ratio is much smaller (Fig. 8c). This decrease in the cloud residuals is almost entirely driven by changes in SA and can either indicate that *during* is influenced by anthropogenic sources, or that SA is formed in the cloud droplets through aqueous-phase oxidation of $SO_2$.

Overall, the chemical composition of the aerosol particles is similar throughout the cloud event. This indicates that the aerosol

665   particles were subject of cloud processing already before we observed this cloud case, or cloud processing occurs on longer time scales than regarded in this study.

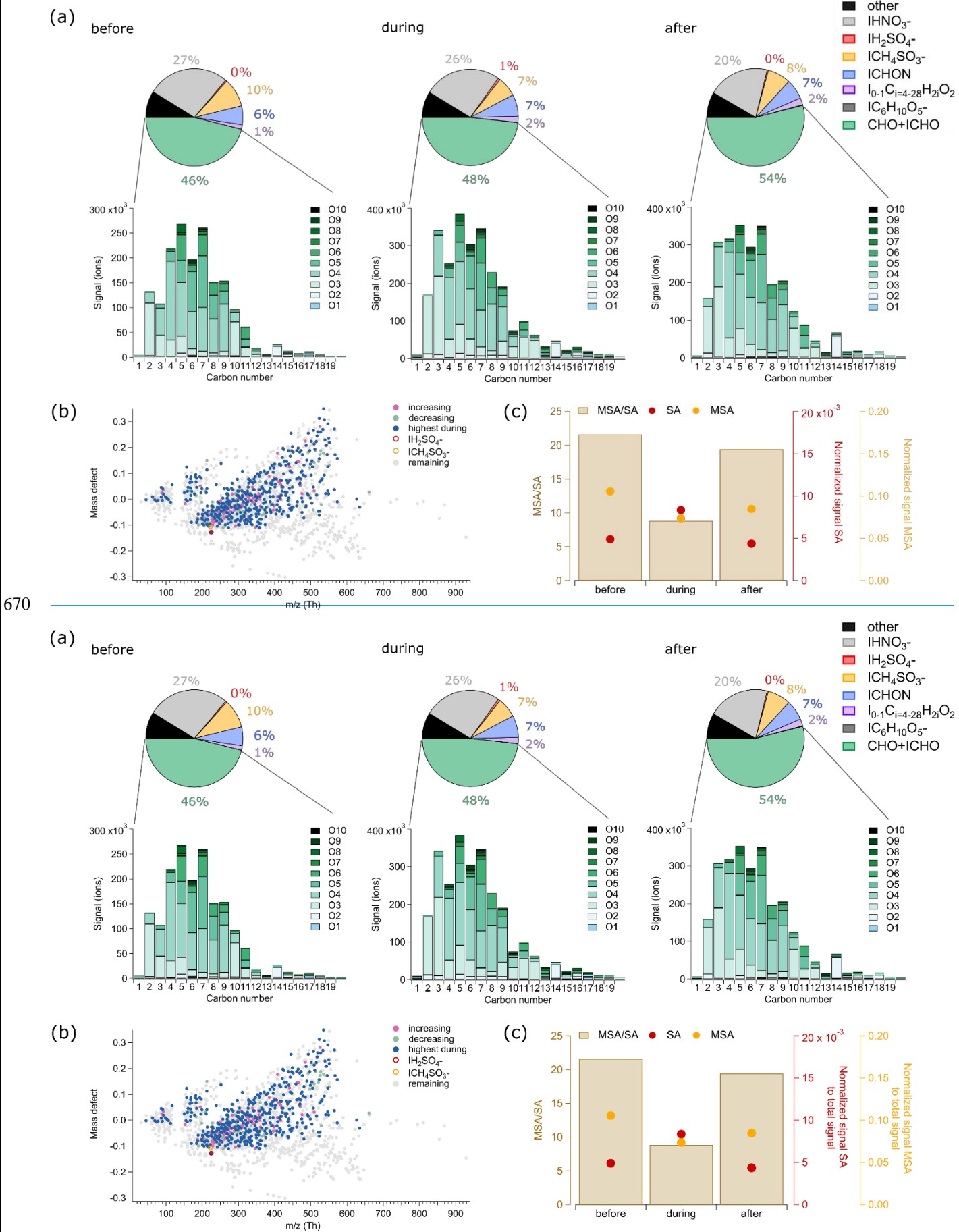

**Figure 8: Cloud case on May 18, 2020, chemical composition of total particles before, and after, of cloud residuals during the cloud event. (a) Pie chart showing the relative contribution of groups and individual compounds to the measured FIGAERO-CIMS signal. Green: CHO+ICHO. Purple: fatty acids, blue: ICHON, yellow: $ICH_4O_3S$- (MSA), red: $IH_2O_4S$- (SA), grey: $IHNO_3$- (NA). Bar chart with absolute signal of CHO+ICHO compounds before, during and after the cloud event grouped by the number of carbon (1-19) and oxygen (O1-O10) atoms. (b) Mass defect highlighting individual compounds that show an increasing or decreasing trend during**

the cloud event (all: all compounds detected with the FIGAERO-CIMS; increasing: compounds of which the relative signal is increasing from before, via during until after the cloud; decreasing: compounds of which the relative signal is decreasing from before, via during until after the cloud; highest during: compounds of which the relative signal is highest during the cloud compared to before and after). (c) Ratio of MSA to sulfuric acid (SA) and their normalized signal to the total signal before, during and after the cloud event.

## 4 Conclusions

In this study we provide the first year-long in situ chemical characterization of cloud residuals at molecular level from the Arctic. We investigated the chemical composition of cloud residuals measured with a FIGAERO-CIMS behind a GCVI inlet at the Zeppelin Observatory, Svalbard, during the NASCENT campaign. From December 2019 until December 2020, we analyzed in total 14 cloud events, two in December (one in 2019 and one in 2020), two in May 2020, eight in June 2020, one in September 2020, and one in October 2020. No cloud residual sample was collected during the haze period.

The number concentrations and size distributions of the cloud residuals agreed roughly with the seasonal pattern observed in the long-term study by Karlsson et al. (2021). We found that the overall chemical composition of the cloud residuals follows the expected annual cycle of aerosol chemical composition in the Arctic, with a large contribution of naturally derived secondary aerosol in the form of MSA during spring and summer. Organic aerosol was present in the cloud residuals during the entire year, with higher absolute signals in the summer compared to winter. Inorganic acids, namely nitric and sulfuric acid, had their largest contribution, both absolute and relative, in spring and late summer, indicating anthropogenic influence. The biomass burning tracer levoglucosan was observed in the summer cloud residuals as well, although its relative contribution to the total measured signal was below 1 % during the entire year. Our results indicate that most of the large enough aerosol particles serve as cloud seeds in the Arctic. In this aerosol-limited environment the chemical properties of the aerosol particles are not as relevant as the physical properties (size and number).

We observed a general relation between the amount of Aitken mode particles and the amount of MSA in the cloud residuals, where higher contributions of Aitken mode particles were present at elevated levels of MSA in the summer. This indicates that during this time of the year, MSA clearly contributes to the growth of newly formed particles into CCN sizes.

In a cloud case study from May 18, 2020 we investigated the change in the aerosol chemical composition due to cloud processing. To do so, we identified district periods before, during and after the cloud event, during which the air was coming from the same region east of Svalbard. This cloud case revealed that after the cloud event, the aerosol population contained a larger relative and absolute signal of oxygenated hydrocarbons (CHO+ICHO), possibly linked to aqueous phase oxidation in the cloud droplets. Nitric acid showed decreasing relative and absolute signal, possibly linked to evaporation. Among individual compounds that showed a relative increase throughout the cloud event are succinic, malonic and glutaric acid, as well as levoglucosan. No clear increase of MSA in the cloud residuals was observed, thus not supporting results from previous studies that indicated MSA to be formed in cloud droplets. However, we note that the observed cloud event was rather short (9 hours). Overall, the presence of the cloud did not seem to change the chemical composition of the aerosol particles much. This indicates, at least for the timescales of a few hours the cloud could be observed, that cloud processing mainly has an impact on the mass of the aerosol particles, and to a lower extend on the relative contribution of individual compounds.

Whereas little change was observed for the chemical composition of the aerosol particles in comparison to the cloud residuals, the presence of the cloud reduced the number of nucleation and Aitken mode particles. This can be attributed to coagulation

of particles and cloud droplets. Overall, the cloud event shifted the average submicron particle population to fewer but slightly larger accumulation mode particles.

The method used in this study allowed us to only investigate the organic fraction and some inorganic acids in the cloud residuals. As the Arctic aerosol also contains other species such as sea salt, heavy metals or dust, our results only reflect part of the cloud residual composition. Nevertheless, our study highlights the importance of natural marine sources as CCN in the Arctic region and further suggests that the entire organic aerosol fraction can serve as CCN in the Arctic.

For future experiments it would be crucial to identify the size-segregated cloud residual chemical composition. Especially of the Aitken mode particles, which play an important role in acting as cloud seeds during a large part of the year (Lawler et al., 2021). In addition, the chemical composition of the total particle composition (cloud residuals and interstitial aerosol particles) along with the cloud residual composition should be investigated – in analogy to the data available for the size distributions. This additional information would shed light on the chemical composition of the interstitial aerosol particles, and allow us to better determine the properties of CCN. Offline and size segregated filter sampling for later FIGAERO-CIMS analysis could be a suitable option.

In the sunlit part of the year, the contribution of natural, local aerosol particle sources to the Arctic aerosol are higher than in the dark period. Our results show that these sources provide relevant seeds for cloud formation in the Arctic. Since the highest cloud cover in Ny-Ålesund is usually expected in summer, the locally produced aerosol particles could be more important for cloud formation than the long-range transported aerosol particles in the dark period. Especially in the context of a warming Arctic, where more open ocean provides a larger source for phytoplankton emissions and thereby also MSA formation, the marine environment could provide an important source of atmospheric aerosol particles able to activate as cloud droplets.

**Data availability**

The data of this study will be available on the Bolin Centre Database (https://bolin.su.se/data/). The meteorological data is available on the EBAS data base (https://ebas-data.nilu.no). The Cloudnet data used in this study are generated by the Aerosol, Clouds and Trace Gases Research Infrastructure (ACTRIS) and are available from the ACTRIS Data Centre using the following links: https://hdl.handle.net/21.12132/1.6ad98e2e244346e8, https://hdl.handle.net/21.12132/1.015016f142ae45b7, https://hdl.handle.net/21.12132/1.2068d2269e014b15, https://hdl.handle.net/21.12132/1.61025d8bee914c66.

**Author contributions**

CM, PZ and RK were responsible for funding and conceptualization. YG, KS, SH and CM performed the FIGAERO-CIMS measurements. YG, KS and SH wrote code for FIGAERO-CIMS analysis. RK, PZ and GF operated the GCVI and provided the GCVI data. GF provided the DMPS data. YG analyzed the FIGAERO-CIMS, and visualized the GCVI, DMPS and FIGAERO-CIMS data. KS visualized the trajectories. YG wrote the manuscript with input from CM. CM was responsible for supervision. All co-authors read and commented on the manuscript.

**Competing interests**

Some of the authors are co-editors at ACP.

**Acknowledgements**

We gratefully acknowledge financial support from the Knut and Alice Wallenberg (KAW) foundation (WAF project CLOUDFORM, grant no. 2017.0165 and ACAS project # 2016.0024). The work was also financially supported by Swedish Environmental Protection Agency (Naturvårdsverket) and by the funding agency FORMAS (IWCAA project # 2016-01427). This work was supported by the Swedish Research Council (Vetenskapsrådet starting grant, project number 2018-05045 and project number 2016-05100). This project has received funding from the European Union's Horizon 2020 research and innovation programme under grant agreement No 821205 (FORCeS). We owe great thanks to Peter Tunved from ACES for harmonizing the DMPS2 data. We are grateful to Kerstin Ebell for help with the interpretation of the Cloudnet classification data. The authors thank NILU, Norway, especially Wenche Aas, Anne Hjellbrekke, and Ove Hermansen for providing the meteorological data (T, RH). We are very thankful for all the support from Helge Tore Markussen and Vera Sklet from the Norwegian Polar Institute in Ny-Ålesund, and from the technicians from the Norwegian Polar Institute in Ny-Ålesund and from ACES, especially, Christer Sørem, Christelle Guesnon, Håkon Jonsson Ruud, Svein-Torgar Oland Paulsen, Filip Heitmann, Ondrej Tesar and Tabea Hennig. We owe great thanks to NILU for providing zero air. We acknowledge ACTRIS and Finnish Meteorological Institute for providing the data set which is available for download from https://cloudnet.fmi.fi/. The cloud radar data for Ny Ålesund was provided by the University of Cologne, the ceilometer and microwave radiometer data by the Alfred Wegener Institute, Helmholtz Centre for Polar and Marine Research. We gratefully acknowledge the funding by the Deutsche Forschungsgemeinschaft DFG (German Research Foundation) - project number 268020496 - TRR 172, within the "Transregional Collaborative Research Center 'ArctiC Amplification: Climate Relevant Atmospheric and SurfaCe Processes, and Feedback Mechanisms (AC)3'".

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

# Supplementary

# Revealing the chemical characteristics of Arctic low-level cloud residuals – in-situ observations from a mountain site

Yvette Gramlich[1,2], Karolina Siegel[1,2,3], Sophie Haslett[1,2], Gabriel Freitas[1,2], Radovan Krejci[1,2], Paul Zieger[1,2], and Claudia Mohr[1,2,a]

[1]Department of Environmental Science, Stockholm University, Stockholm, Sweden
[2]Bolin Centre for Climate Research, Stockholm University, Stockholm, Sweden
[3]Department of Meteorology, Stockholm University, Stockholm, Sweden
[a]now at Department of Environmental System Science, ETH Zurich and Laboratory of Atmospheric Chemistry, Paul Scherrer Institute, Villigen, Switzerland

*Correspondence to*: Claudia Mohr (claudia.mohr@psi.che̶l̶a̶u̶d̶i̶a̶.̶m̶o̶h̶r̶@̶a̶c̶e̶s̶.̶s̶u̶.̶s̶e̶)

## S1 Overview of previous studies on the chemical composition of cloud residuals

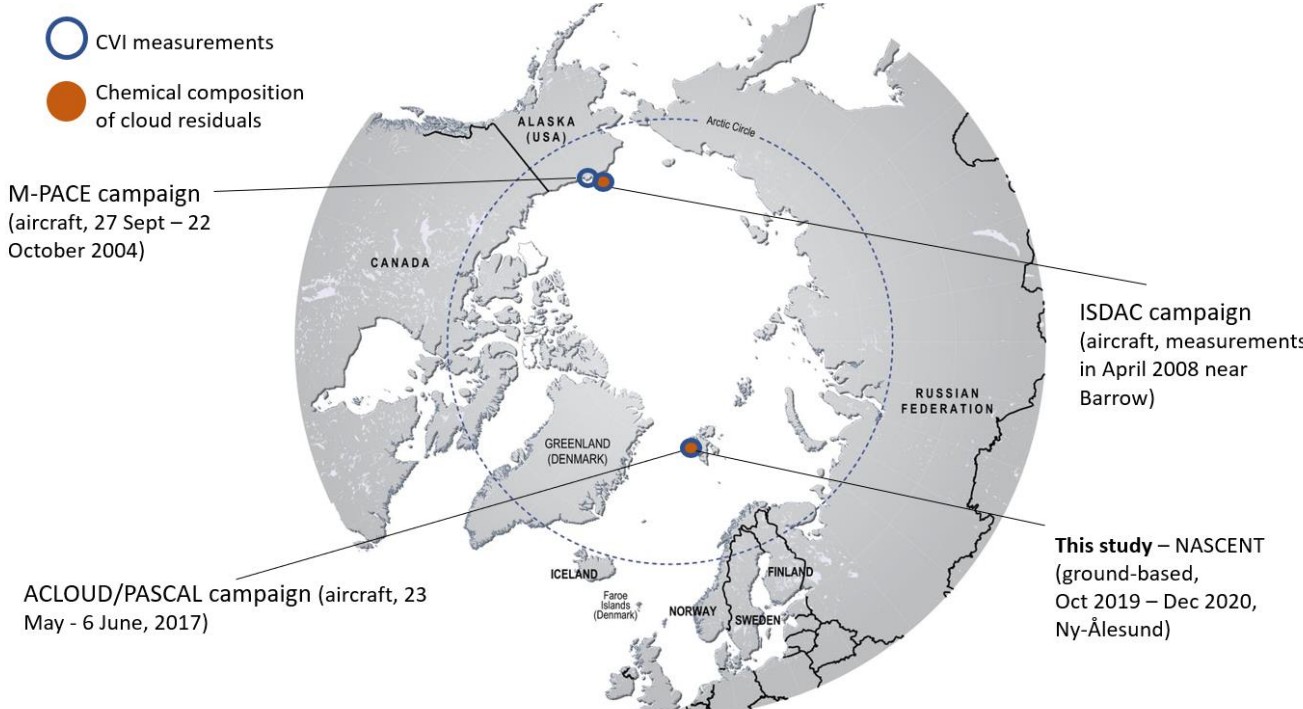

**Figure S1: Map of the Arctic showing the sampling location of NASCENT and the locations of previous studies on the chemical composition of Arctic cloud residuals. (Map taken from https://www.grida.no/resources/8378.) We note that our cloud residual**

samples were obtained using a Ground-based Counterflow Virtual Impactor (GCVI), whereas previous cloud residual samples were aircraft based (CVI) (Verlinde et al., 2007; McFarquhar et al., 2011; Wendisch et al., 2019).

## S2 Background correction for FIGAERO-CIMS data

The background correction was done following the approach recommended in (Cai et al., (2023). We scaled the blank heating signal to the end of the sample signal and subtracted the integrated scaled blank signal from the integrated sample heating (Fig. S2). Since we did not have a blank for each cloud residual sample, we took the blank that was measured closest (with respect to time) to the sample as the respective background (Table S1).

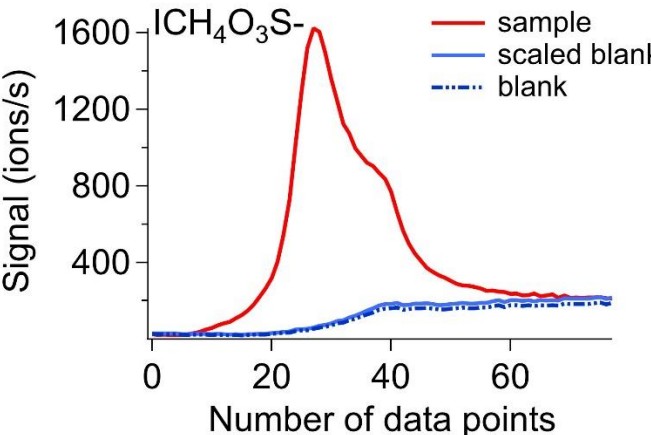

**Figure S2: Example of the signal of CH₄O₃S (methanesulfonic acid, MSA) measured as iodide cluster in the cloud residual of Jun 26, 2020, showing how the scaled blank signal was used for background determination. The dashed blue line is the original blank signal and the solid blue line the blank signal scaled to the end of the sample heating. The interval between two data points equals a time period of 30 s.**

**Table S1: Cloud residual samples and the respective blanks used for the background correction. The time in the brackets refers to the respective FIGAERO-CIMS sampling time of the blanks. The sampling times for the cloud residual samples can be found in the main text, Table 1.**

| Cloud residual sample | Blank used | Cloud residual sample | Blank used |
|---|---|---|---|
| Dec 25, 2019 | Jan 19, 2020 (13:05:51-15:36:14) | Jun 25, 2020 | Jun 25, 2020 (13:29:00-15:59:09) |
| May 18, 2020 | May 18, 2020 (11:03:03-13:33:05) | Jun 26, 2020 | Jun 26, 2020 (10:29:00-12:59:08) |
| May 21, 2020 | May 21, 2020 (16:17:47-18:47:49) | Jun 27-1, 2020 | Jun 27, 2020 (07:28:59-09:59:07) |
| Jun 2, 2020 | Jun 12, 2020 (14:50:11-17:20:19) | Jun 27-2, 2020 | Jun 27, 2020 (07:28:59-09:59:07) |
| Jun 12-1, 2020 | Jun 12, 2020 (14:50:11-17:20:19) | Sep 12, 2020 | Sep 12, 2020 (23:01:39-Sep 13 01:32:00) |
| Jun 12-2, 2020 | Jun 12, 2020 (14:50:11-17:20:19) | Oct 28, 2020 | Sep 12, 2020 (23:01:39-Sep 13 01:32:00) |

| Jun 14, 2020 | Jun 13, 2020 (22:20:25--Jun 14 00:50:03) | Dec 9, 2020 | Sep 12, 2020 (23:01:39-Sep 13 01:32:00) |
|---|---|---|---|

## S3 Cloud residual size distributions

The number size distributions of all cloud residuals not shown in the main text are presented in Figure S3.

During the times when we sampled the cloud residuals on June 25, June 27-1 and September 12, 2020, there was also drizzle present. This can be seen in data from the condensation particle counter (CPC, model 3772, TSI Inc., USA), as well as in the Cloudnet target classification (Fig. S4). The drizzle droplets can splash when they hit the funnel of the wind tunnel of the GCVI and produce several, smaller droplets. This can then be seen as a spike in the total particle number concentration of the CPC measured at a time resolution of 1 s ($N_{tot}$ 1s). This concentration can be compared to the particle number concentration

measured with another CPC (model 3772, TSI Inc., USA) behind the differential mobility analyser (DMA, medium Vienna-type, length 0.28 m, outer radius 0.033 m, inner radius 0.025 m) and integrated over the entire size range ($N_{int}$, time resolution 7 min) by averaging it to the same time resolution ($N_{tot}$ mean). If the size selected by the DMA ($D_{scan}$) is in the size range of the Aitken mode particles when droplet splashing occurs, the integrated number concentration ($N_{int}$) will be much higher than $N_{tot}$ mean for the same time interval. Additionally, a large number of Aitken mode particles can be observed in the number

size distribution. Therefore, we removed the datapoints for the number size distributions where the median ratio of $N_{int}/N_{tot}$ over the entire 2.5 h sampling time was larger than a certain threshold. Fig. S6 shows an example (from September 12, 2020) of how drizzle splashing can be observed in the Differential Mobility Particle Sizer (DMPS) data. The median ratios, the selected threshold and the corresponding number of datapoints that are removed are shown in Table S2.


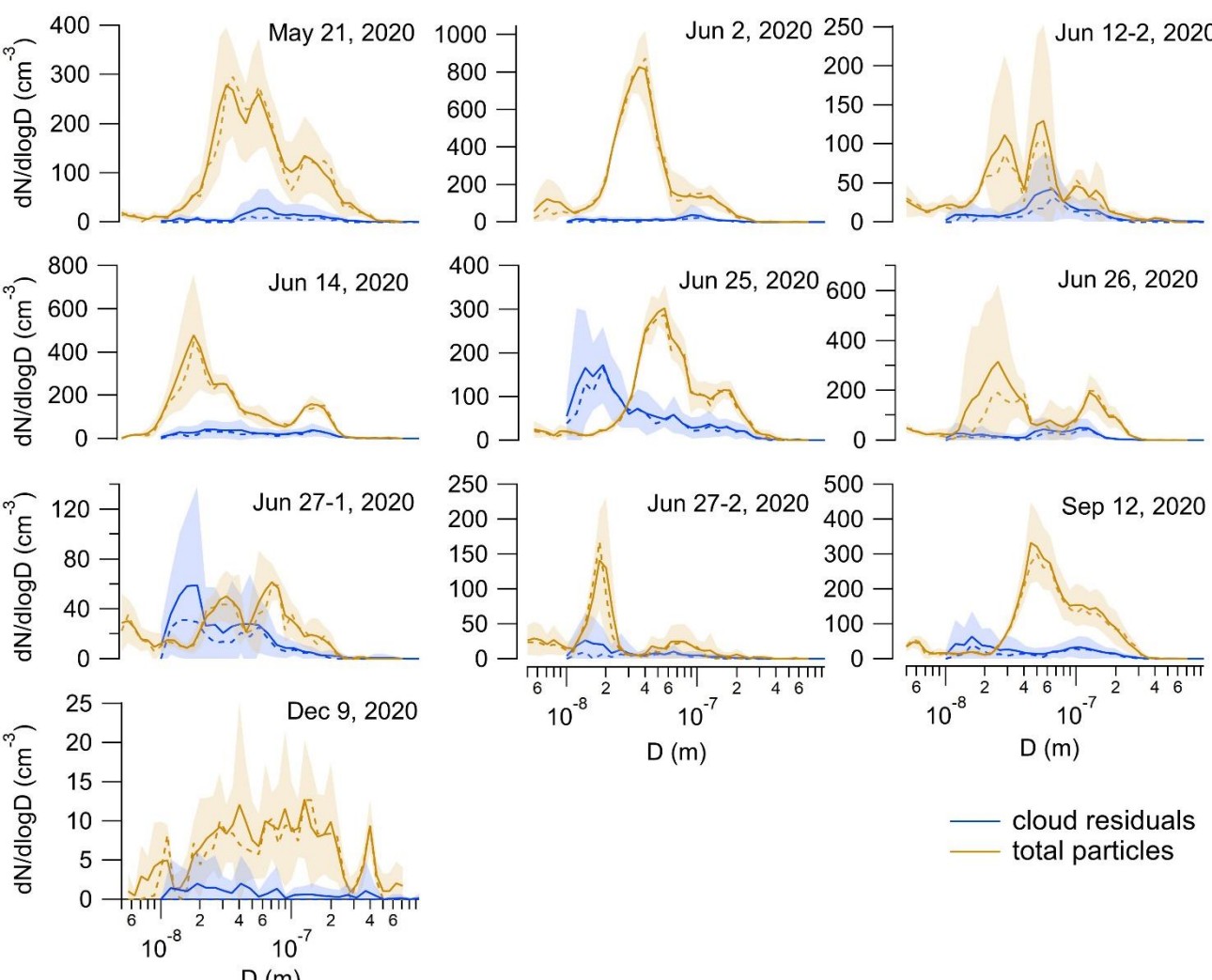

**Figure S3:** Average number size distributions of the cloud residuals and the total particles during the corresponding 2.5 h FIGAERO-CIMS sampling time of all remaining samples not shown in the main text. The shaded area represents the standard deviation.

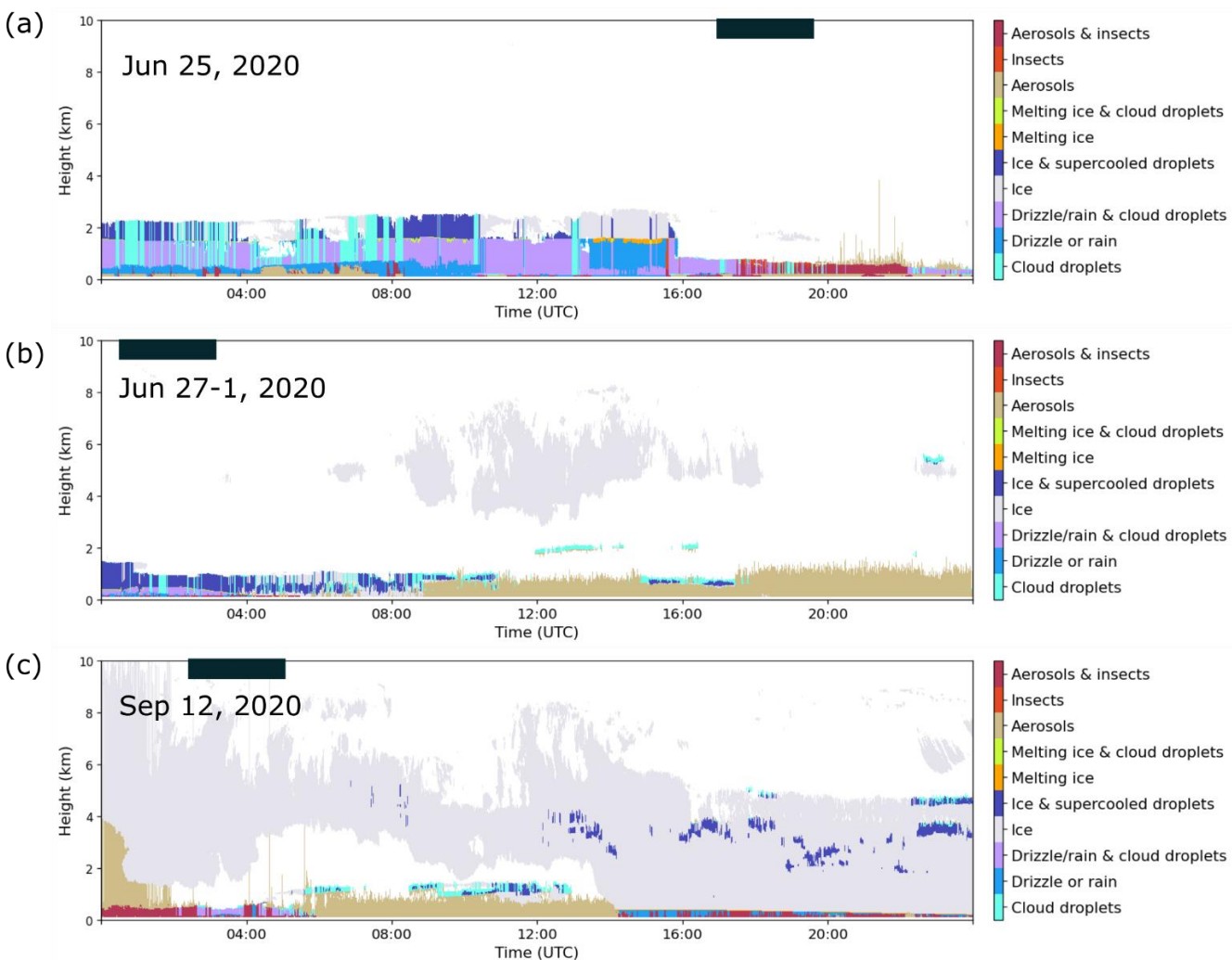

**Figure S4: Cloudnet target classification (modified by adding the respective cloud residual sample date, and indicating the approximate sampling times of the cloud with a black bar on top of each subfigure) for (a) June 25, 2020 (Cloudnet (2021), last access: 2022-11-24, 15:35 UTC), (b) June 27-2, 2020 (Cloudnet (2021b), last access: 2022-11-24, 15:28 UTC), (c) September 12, 2020 (Cloudnet (2021a), last access: 2022-11-24, 15:38 UTC), indicating that there was drizzle present during the sampling times of the cloud residuals.**


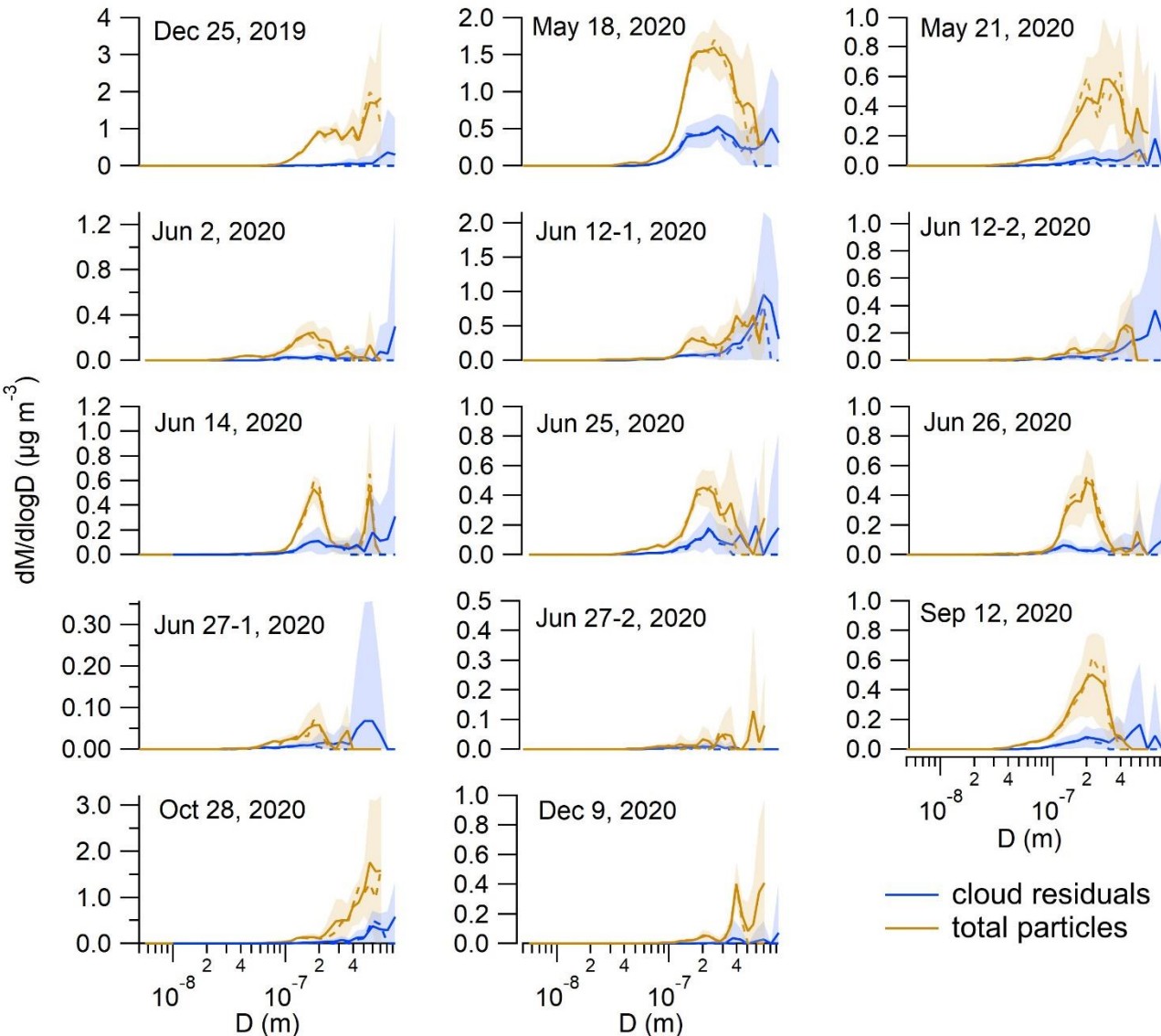

**Figure S5: Average mass size distributions of all the cloud residual samples and the total particle population. The shaded area indicates the standard deviation. For the conversion from number to mass a density of 1.3 g cm$^{-3}$ was used, representing secondary organic aerosol (e.g. Alfarra et al., 2006; Malloy et al., 2009).**

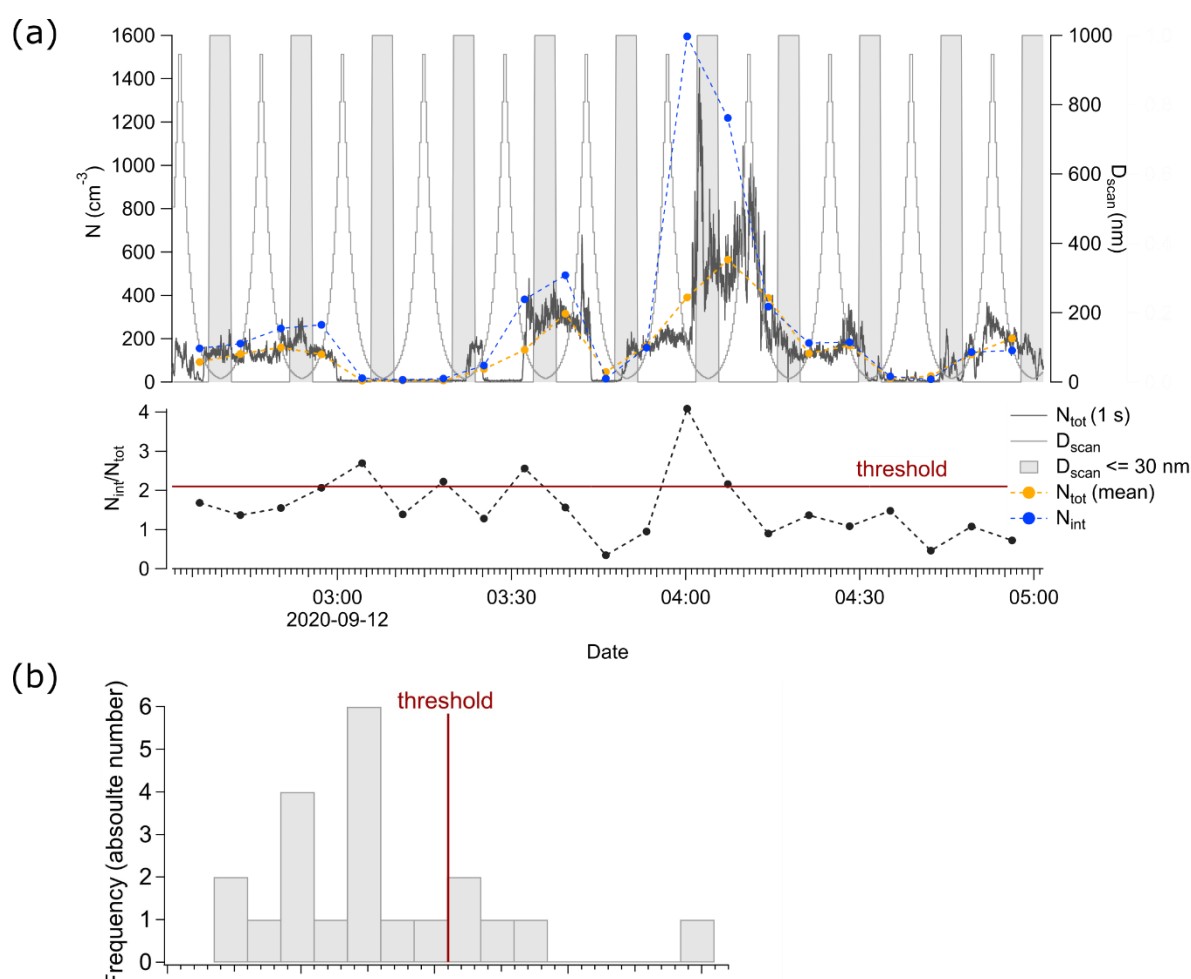

 **Figure S6: (a) DMPS data from the cloud residual sample of September 12, 2020, as an example of how the droplet splashing of drizzle droplets can be seen as spikes in the $N_{tot}$ (1 s) time series. $D_{scan} \le 30$ indicates the time periods when the size selected by the DMA was in the small size range, selecting particles up to diameters of 30 nm. The lower panel shows the ratio of $N_{int}/N_{tot}$ and the location of the threshold. (b) Histogram of the ratio $N_{tot}/N_{int}$ indicating the location of the selected threshold for filtering the data. For the ratio $N_{tot}/N_{int}$ we took $N_{tot}$ (mean).**

**Table S2: Cloud case, median ratio of $N_{int}/N_{tot}$, selected threshold above which the datapoints were removed, and the corresponding number of datapoints that are removed ($Num_{removed}$) from the total number of datatpoints ($Num_{total}$).**

| Cloud case | Median ratio $N_{int}/N_{tot}$ | Threshold | $Num_{removed}/Num_{total}$ |
|---|---|---|---|
| Jun 25, 2020 | 1.5 | 1.6 | 9/22 |
| Jun 27-1, 2020 | 1.2 | 1.9 | 4/22 |
| Sep 12, 2020 | 1.4 | 2.1 | 5/21 |

## S4 Signal of formic acid during the heating

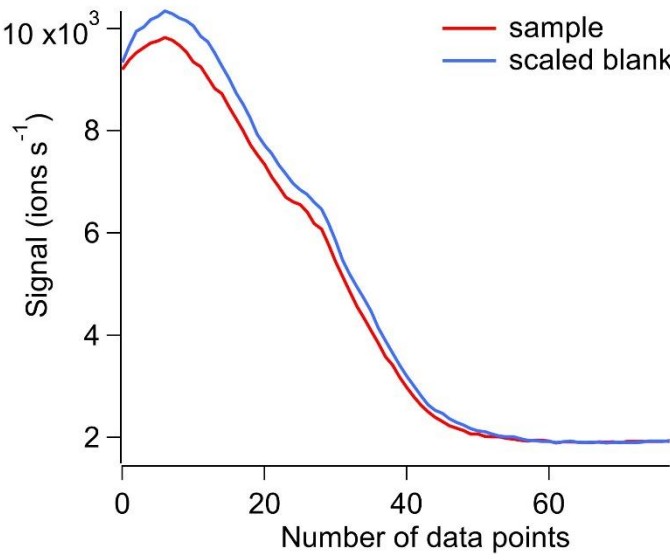


**Figure S7: Signal of formic acid (ICH₂O₂-) during the heating of the sample and the scaled blank, respectively, as a function of heating time. The interval between two data points equals a time period of 30 s. As an example, the signal here is presented from the cloud residual sample on Jun 25, 2020.**



## S5 Mass spectra of cloud residuals

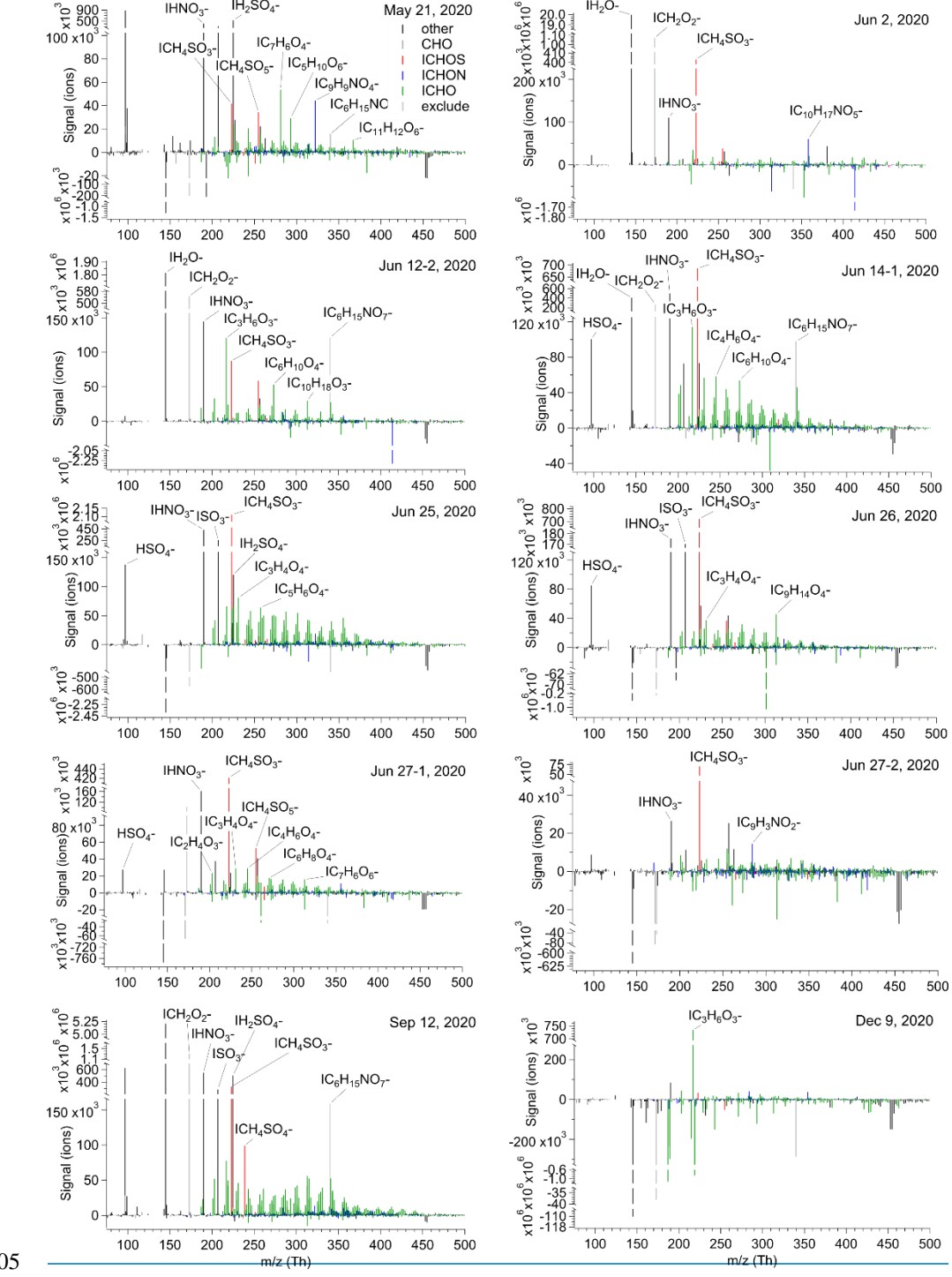

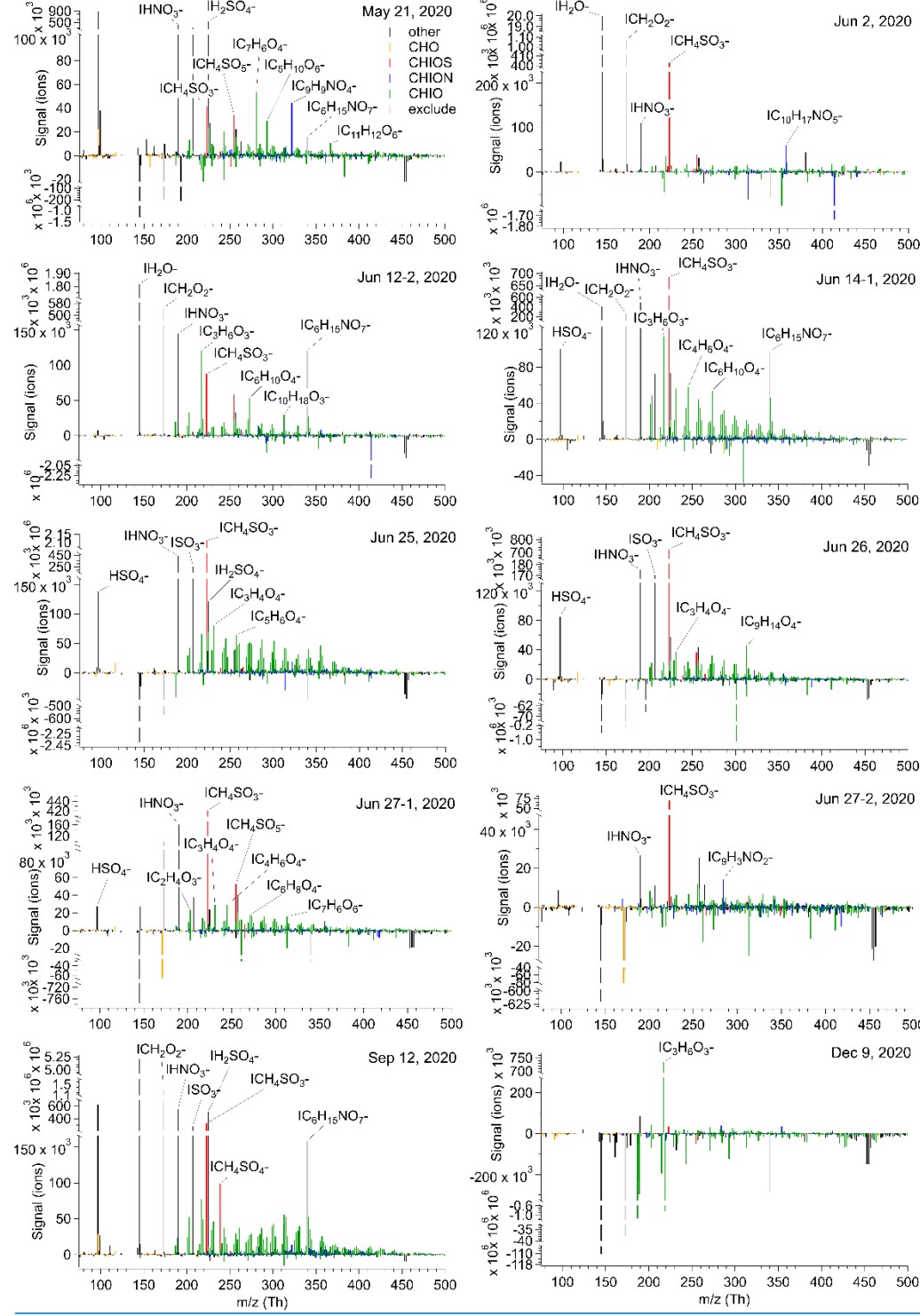

**Figure S8: Mass spectra of all cloud residual samples not shown in the main text.**

### S6 Chemical composition of cloud residuals

The chemical formulas referring to lactic acid ($IC_3H_6O_3-$), palmitic acid ($IC_{16}H_{32}O_2-$) and stearic acid ($IC_{18}H_{36}O_2-$) could potentially be related to handling of the GCVI. Based on our data, we were not able to clearly identify if they are only a background signal or if they are actual compounds in the cloud residuals. We observe especially high signals after background subtraction of the compounds in question during the times when the gap in time of sampling and taking a blank was the highest (Dec 25, 2019; Dec 9, 2020, Fig. S9). Fig. S10 shows the chemical composition of the cloud residuals when excluding lactic, palmitic and stearic acid from the absolute and relative signal.

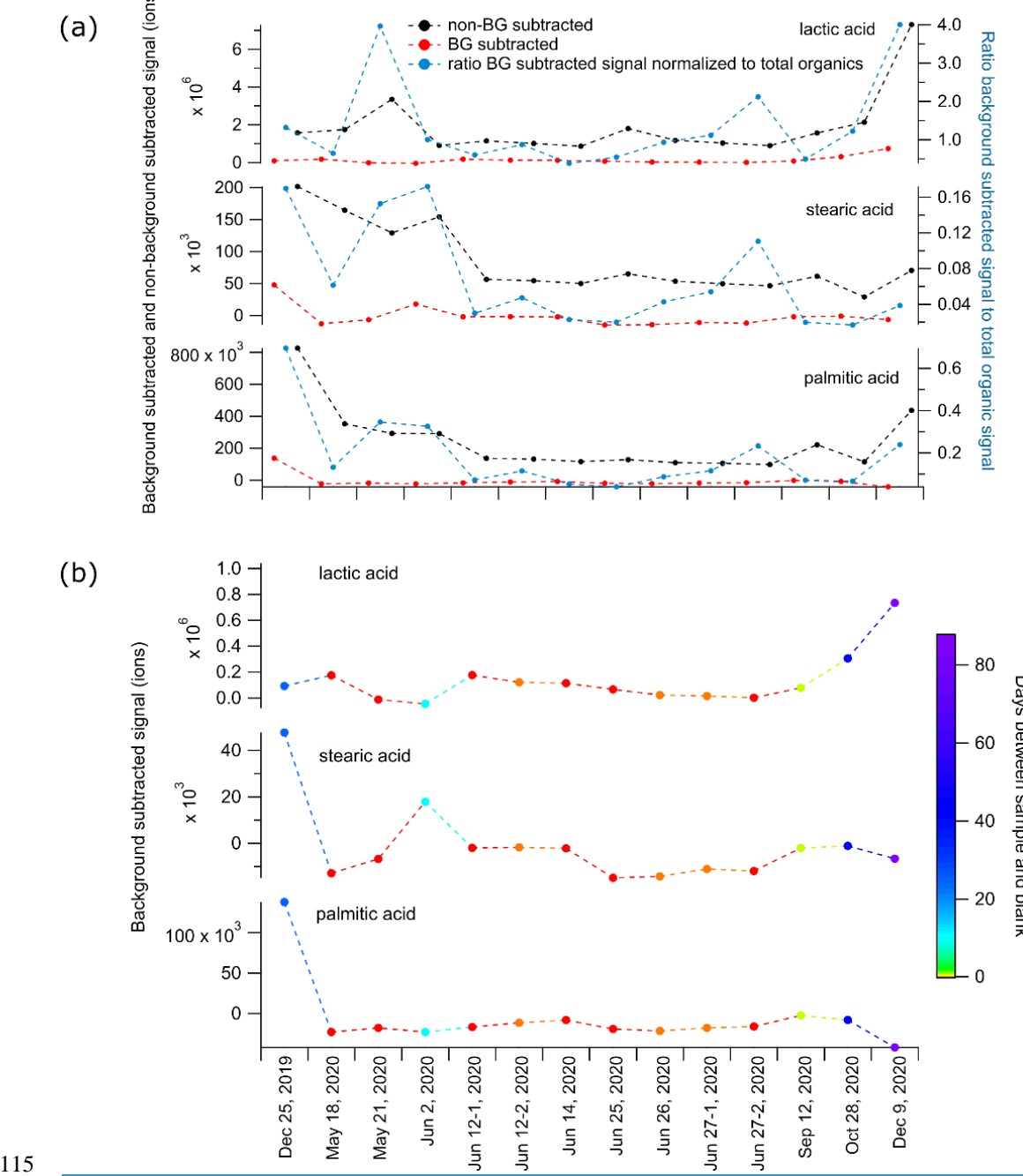

**Figure S9: (a) Background (BG) subtracted and non-BG subtracted signals of lactic, stearic and palmitic acid and the ratio of their BG subtracted signal to the total organic signal. (b) Background subtracted signals of lactic, stearic and palmitic acid color coded by the time difference between the sample and the blank.**


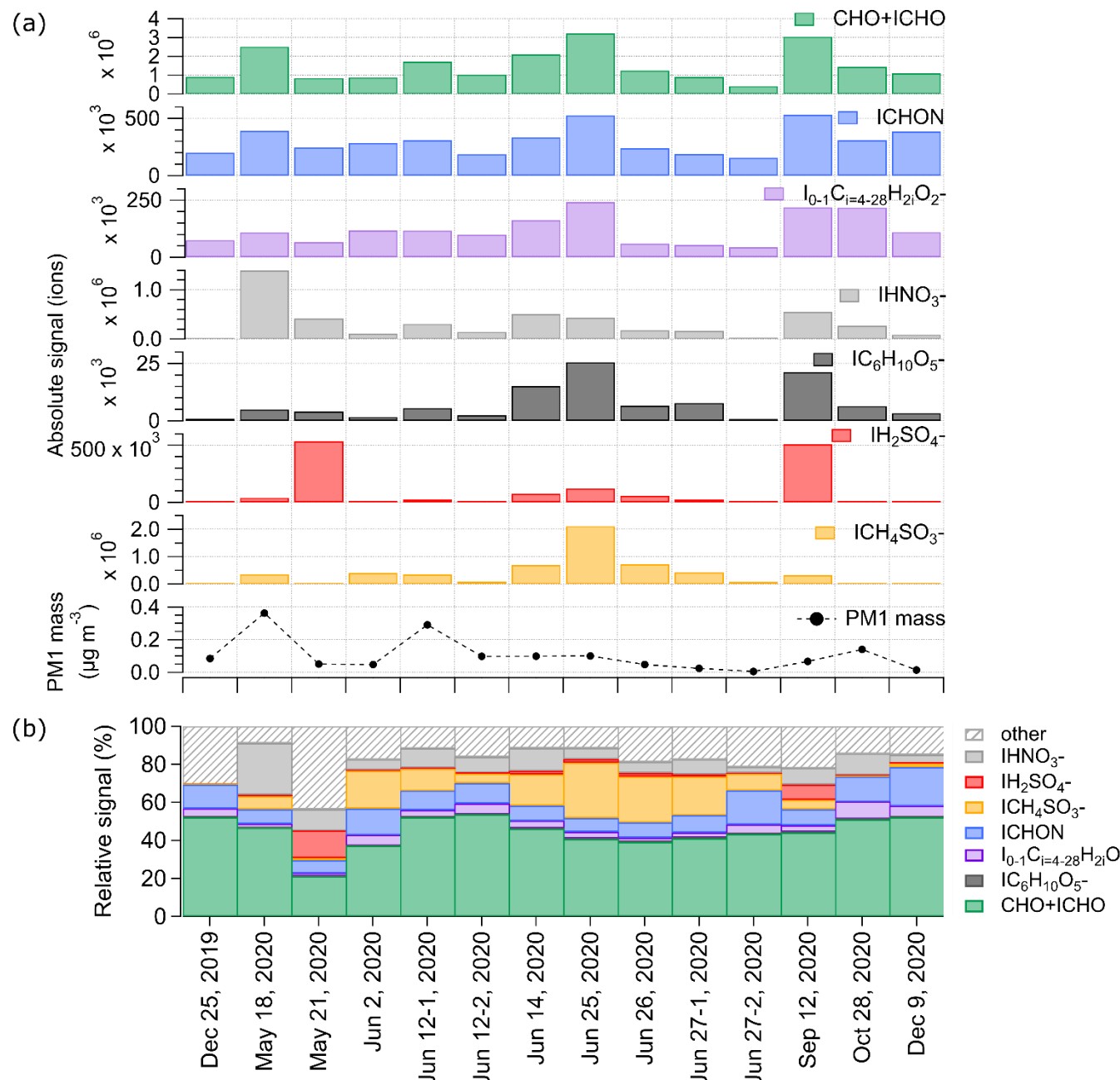

Figure S10: (a) Absolute signal of different compound groups (CHO+ICHO, ICHON, $I_{0-1}C_{i=4-28}H_{2i}O_2-$ (fatty acids), $IHNO_3-$ (NA $IC_6H_{10}O_5-$ (levoglucosan), $IH_2SO_4-$ (SA), $ICH_4SO_3-$ (MSA)) in the different cloud residual samples, and the respective $PM_1$ mass. (b) Relative signal of different compound groups in the different cloud residual samples. Note: in the absolute signal view in (a) the CHO+ICHO group contains also the signal of $IC_6H_{10}O_5-$, and $I_{0-1}C_{i=4-28}H_{2i}O_2-$, whereas for the relative signal in (b) the signal from these two groups have been subtracted from CHO+ICHO. This figure is similar to Fig. 5 from the main text, but excluding the three compounds that might be linked to hygiene products (lactic, palmitic and stearic acid).

## S76 Back trajectories

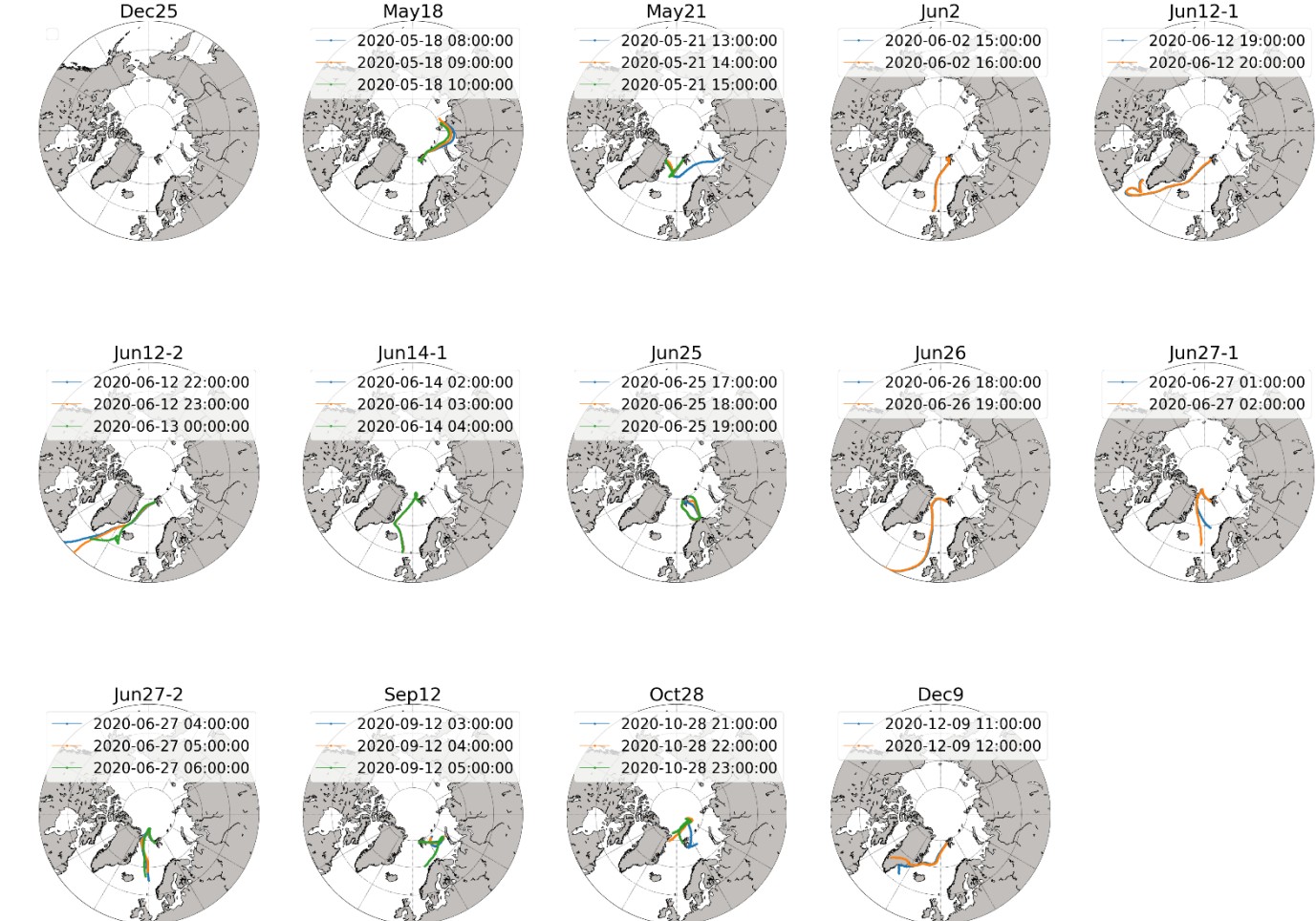

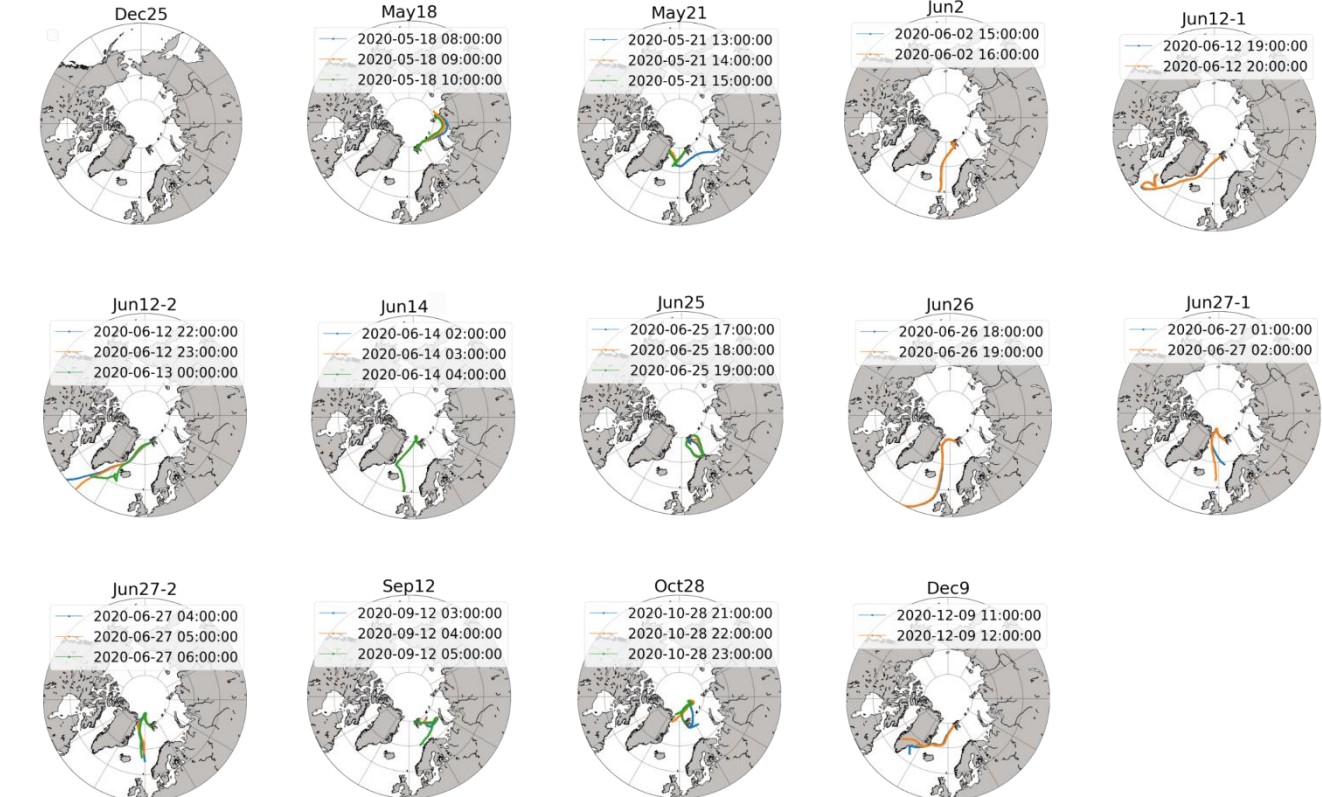

**Figure S119: HYSPLIT 5-day back trajectories for the sampling times of all the cloud residual samples.**



**S8<s>7</s> Cloud case May 18, 2020**

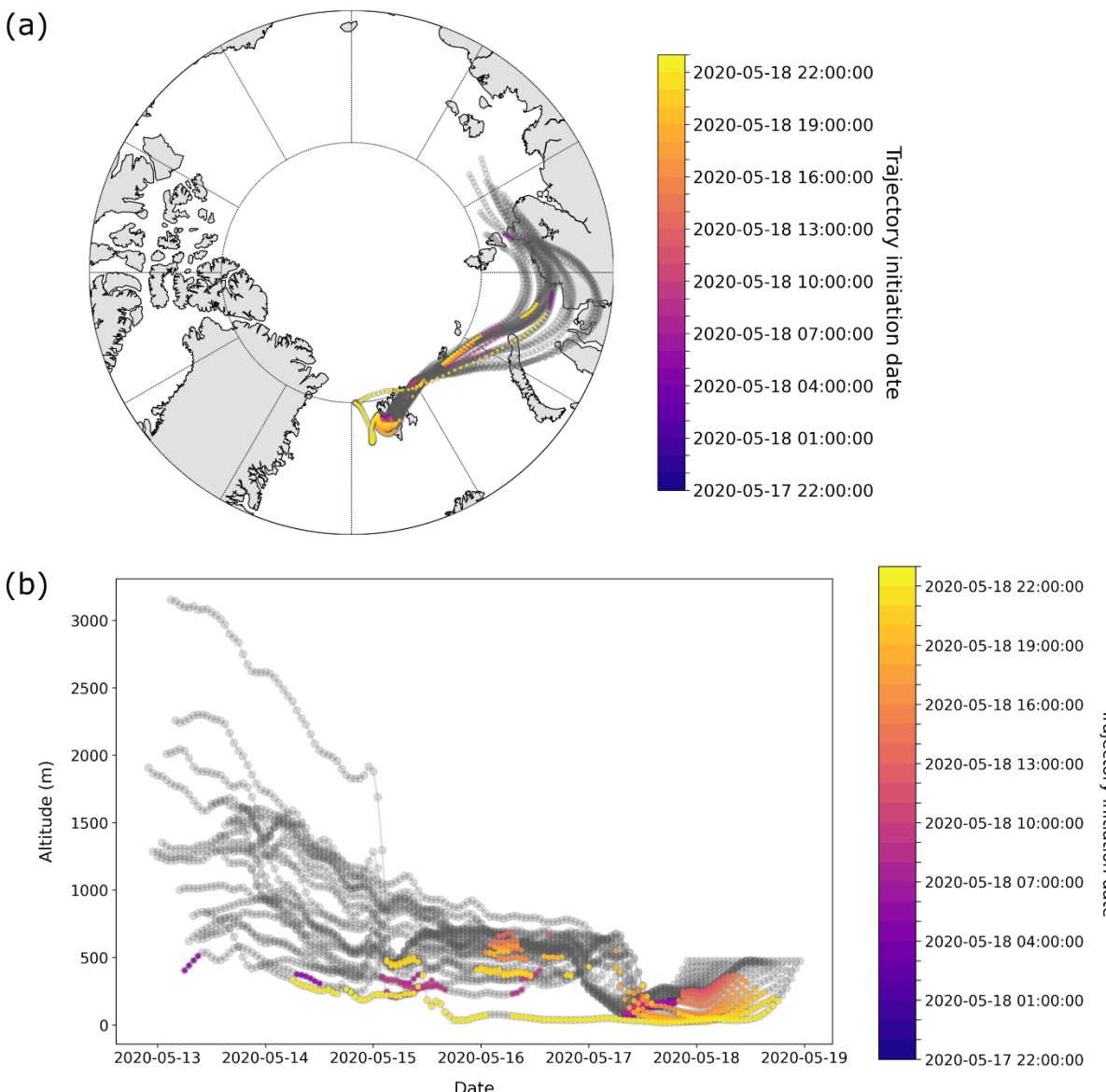

Figure S12<s>0</s>: 5 days back trajectories of air masses arriving at the Zeppelin Observatory before, during and after the cloud event color coded by time and height with respect to the boundary layer height (BL). (a) Map view, (b) Trajectory height as a function of time. Grey colors indicate times above the BL, and colors indicate times below the BL.


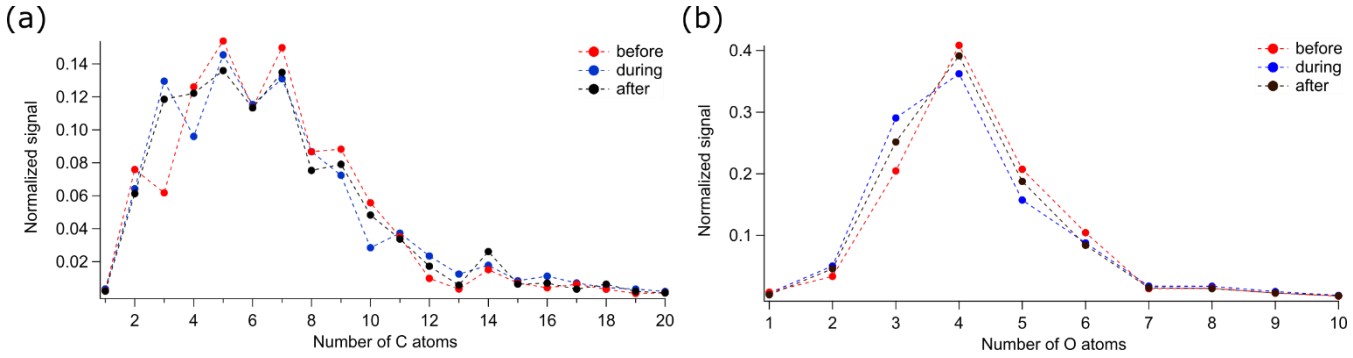

**Figure S131: Signal contributions of compounds grouped according to different numbers of (a) carbon and (b) and oxygen to the total CHO+CHOI signal for the cloud case May 18, 2020.**

## S References

Alfarra, M. R., Paulsen, D., Gysel, M., Garforth, A. A., Dommen, J., Prévôt, A. S. H., Worsnop, D. R., Baltensperger, U., and Coe, H.: A mass spectrometric study of secondary organic aerosols formed from the photooxidation of anthropogenic and biogenic precursors in a reaction chamber, Atmospheric Chem. Phys., 6, 5279–5293, https://doi.org/10.5194/acp-6-5279-2006, 2006.

Cai, J., Daellenbach, K. R., Wu, C., Zheng, Y., Zheng, F., Du, W., Haslett, S. L., Chen, Q., Kulmala, M., and Mohr, C.: Characterization of offline analysis of particulate matter with FIGAERO-CIMS, Atmospheric Meas. Tech., 16, 1147–1165, https://doi.org/10.5194/amt-16-1147-2023, 2023.

Cloudnet: Classification data from Ny-Ålesund on 12 September 2020, 2021a.

Cloudnet: Classification data from Ny-Ålesund on 25 June 2020, 2021b.

Cloudnet: Classification data from Ny-Ålesund on 27 June 2020, 2021c.

Malloy, Q. G. J., Nakao, S., Qi, L., Austin, R., Stothers, C., Hagino, H., and Cocker, D. R.: Real-Time Aerosol Density Determination Utilizing a Modified Scanning Mobility Particle Sizer—Aerosol Particle Mass Analyzer System, Aerosol Sci. Technol., 43, 673–678, https://doi.org/10.1080/02786820902832960, 2009.

McFarquhar, G. M., Ghan, S., Verlinde, J., Korolev, A., Strapp, J. W., Schmid, B., Tomlinson, J. M., Wolde, M., Brooks, S. D., Cziczo, D., Dubey, M. K., Fan, J., Flynn, C., Gultepe, I., Hubbe, J., Gilles, M. K., Laskin, A., Lawson, P., Leaitch, W. R., Liu, P., Liu, X., Lubin, D., Mazzoleni, C., Macdonald, A.-M., Moffet, R. C., Morrison, H., Ovchinnikov, M., Shupe, M. D., Turner, D. D., Xie, S., Zelenyuk, A., Bae, K., Freer, M., and Glen, A.: Indirect and Semi-direct Aerosol Campaign: The Impact of Arctic Aerosols on Clouds, Bull. Am. Meteorol. Soc., 92, 183–201, https://doi.org/10.1175/2010BAMS2935.1, 2011.

Verlinde, J., Harrington, J. Y., McFarquhar, G. M., Yannuzzi, V. T., Avramov, A., Greenberg, S., Johnson, N., Zhang, G., Poellot, M. R., Mather, J. H., Turner, D. D., Eloranta, E. W., Zak, B. D., Prenni, A. J., Daniel, J. S., Kok, G. L., Tobin, D. C., Holz, R., Sassen, K., Spangenberg, D., Minnis, P., Tooman, T. P., Ivey, M. D., Richardson, S. J., Bahrmann, C. P., Shupe, M., DeMott, P. J., Heymsfield, A. J., and Schofield, R.: The Mixed-Phase Arctic Cloud Experiment, Bull. Am. Meteorol. Soc., 88, 205–222, https://doi.org/10.1175/BAMS-88-2-205, 2007.

Wendisch, M., Macke, A., Ehrlich, A., Lüpkes, C., Mech, M., Chechin, D., Dethloff, K., Velasco, C. B., Bozem, H., Brückner, M., Clemen, H.-C., Crewell, S., Donth, T., Dupuy, R., Ebell, K., Egerer, U., Engelmann, R., Engler, C., Eppers, O., Gehrmann, M., Gong, X., Gottschalk, M., Gourbeyre, C., Griesche, H., Hartmann, J., Hartmann, M., Heinold, B., Herber, A., Herrmann, H., Heygster, G., Hoor, P., Jafariserajehlou, S., Jäkel, E., Järvinen, E., Jourdan, O., Kästner, U., Kecorius, S., Knudsen, E. M.,
180   Köllner, F., Kretzschmar, J., Lelli, L., Leroy, D., Maturilli, M., Mei, L., Mertes, S., Mioche, G., Neuber, R., Nicolaus, M., Nomokonova, T., Notholt, J., Palm, M., van Pinxteren, M., Quaas, J., Richter, P., Ruiz-Donoso, E., Schäfer, M., Schmieder, K., Schnaiter, M., Schneider, J., Schwarzenböck, A., Seifert, P., Shupe, M. D., Siebert, H., Spreen, G., Stapf, J., Stratmann, F., Vogl, T., Welti, A., Wex, H., Wiedensohler, A., Zanatta, M., and Zeppenfeld, S.: The Arctic Cloud Puzzle: Using ACLOUD/PASCAL Multiplatform Observations to Unravel the Role of Clouds and Aerosol Particles in Arctic Amplification,
185   Bull. Am. Meteorol. Soc., 100, 841–871, https://doi.org/10.1175/BAMS-D-18-0072.1, 2019.