# Peer review of "Revealing the chemical characteristics of Arctic low-level cloud residuals – in situ observations from a mountain site"

_EGUsphere, 2022_

## Author Response (AR1)

**RC1:**

This paper reports the chemical composition of cloud residuals from the mountain-top field observatory in the European sector of the Arctic, where mixed-phase clouds are frequently observed. The authors' research approach of employing a combination of a counterflow virtual impactor and an advanced chemical ionization mass spectrometer is properly discussed, and the authors also appropriately discuss the limitation of their techniques (e.g., L307-320; L648-658). The results of a yearlong observation/analysis by the authors for their study period/location suggest that the composition of cloud residuals is not substantially unique as compared to general aerosol composition (especially organics), and thereby, the size may be a dominant factor for defining the particle's ability to serve as cloud seeds (unless otherwise other physical properties, such as mixing state, or compositions that are not detectable by FIGAERO-CIMS etc.). Overall, the manuscript is well-written. The findings of this study are worth publishing and being shared amongst the research community. This reviewer supports the publication of this paper in ACP after minor and technical revisions.

We thank the reviewer for their positive assessment of our manuscript. We appreciate the time the reviewer has taken to help improve our work. In the following we respond to each of the reviewer's comments (in bold) individually (normal font). Line numbers in the responses refer to the revised document.

**[Minor comments]**

**L121-124: The reader, who is not familiar with the visibility measurement, would appreciate a bit more general description of how the visibility is measured and how the 1000 m visibility threshold is determined. The reviewer is aware that the authors cite Karlsson et al. later on; however, the current statement seems too concise to understand how valid this threshold is.**

The authors thank the reviewer for indicating this. The authors removed the statement "(visibility < 1000 m for at least 5 min)" from Sect. 2.2, and added more information about the visibility in Sect. 2.3 instead. For that the authors modified L158-159 to "As an indication of cloudy conditions, the visibility sensor (Belfort, Model 6400) that comes with the GCVI was used", and L159-161 to "Similar to Karlsson et al. (2021), we used a visibility threshold of 1000 m, where visibility below this threshold meant presence of a cloud according to the World Meteorological Organization's definition of fog (World Meteorological Organization, 2008)."

**L204-213: This part seems a better fit in Sect. 2.3 as it is explicitly about GCVI.**

Thank you for pointing this out. We moved L204-211 of the first version of the manuscript to Sect. 2.3 (now L149-156) and kept in Sect. 2.5 "All the cloud residual number size distributions in this study were corrected by a factor $k$ (Eq. 1), which is the inverse of the product of the sampling efficiency and the enrichment factor of the GCVI (see Sect. 2.3). No other particle loss calculations were applied. The data for the number size distributions of some samples was filtered for potential droplet splashing according to the procedure described in the supplementary (Sect. S3)."

**L211: All FIGAERO-CIMS signals for cloud residuals (e.g., Fig. 8 during) were also corrected by k?**

No, the FIGAERO-CIMS signals for the cloud residuals were not corrected by $k$. It is only the DMPS number size distributions behind the GCVI that were corrected by $k$. To make this more clear for the reader, the authors made a small addition to L252-253: "The mass-based measurements (e.g. FIGAERO-CIMS) were not corrected by $k$, as …[]".

**Table 1: Please define whether the given time stamps are in UTC or local time. More importantly, the reviewer suggests that the authors add another column listing the ratio of counterflow to input flow (refer to Fig. 5 of Kulkarni et al., 2011; DOI: 10.1080/02786826.2010.539291). Doing this might clarify how the GCVI was operated to the reader without adding too much text.**

The authors added to the header of Table 1 that the given time stamps are in UTC ("Cloud residual samples and the respective sampling start and end date and time (in UTC) of the FIGAERO-CIMS"). The counterflow was always one liter higher than the sample flow, with the counterflow being 16 L/min and the sample flow 15 L/min. Frequent zero checks were done balancing the flows by setting the counterflow close to zero during cloud free conditions. To make this also clear to the reader the authors added this information to L142-146: "This lower cut size of the GCVI was achieved by keeping the counterflow always one liter higher than the sampling flow, 16 and 15 L min$^{-1}$, respectively. Regular zero checks were done during operation where the counterflow was set close to 0 L min$^{-1}$ during cloud free conditions to ensure that the flow control operates well. The quality of counterflow was also routinely tested by switching off the wind tunnel and sampling only counterflow through the sample line."

**L464-465: The authors state that a high Aitken-mode particle fraction coincides with a high MSA signal. This seems not the case for the June 27-2 period as the MSA signal of June 27-2 is at the level of May 21 and other winter periods. Perhaps, rephrasing this part to "… with the MSA/SA of >10 while detecting absolute MSA signal of XX ions" would more precisely reflect what Fig. 6 infers?**

We thank the reviewer for this observation. As suggested by the reviewer, the authors changed L519-520 to "In general, the summer period with a high number contribution of Aitken-mode particles in the cloud residuals coincides with MSA/SA ratios > 10 and absolute MSA signals between around 75.000 and 200.000 ions in the cloud residuals (Fig. 6b)."

**Fig. 7a, L25-26, L587-589: The air masses did not get impacted by precipitation and wet-deposition of aerosols during 5-day backward trajectory history? In other words, does accounting for precipitation change the air mass origin? The reviewer is asking this since some studies set the rainfall thresh (e.g., 7 mm in Gong et al. (2020)) and, if necessary, reclassify the air mass origin to account for potential wash-out of aerosols in tracked air mass (Gong et al., 2020; DOI: 10.5194/acp-20-1431-2020).**

The accumulated precipitation that occurs along all the 5 day back trajectories is below 7 mm, which is the threshold used by Gong et al. (2020) to account for wet removal along the transport way. The sum of precipitation along the back trajectories span a range between 0 and 1 mm, with a median of 0.2 and a mean of 0.3 mm. Since the accumulated precipitation along the trajectories are well below the threshold of 7 mm used by a previous study (Gong et al., 2020), the authors decided to keep the trajectory classification as it is. The authors added L552-553: "The accumulated precipitation along the back trajectories was on average 0.3 mm (median 0.2 mm, max. 1 mm, min. 0 mm)."

**L575-577: While the authors are using careful language here, the reviewer is a bit skeptical about the statement regarding the contribution of lactic acid to INP. Are there any previous studies vilifying lactic acid can act as INP? As the authors might know, the population of atmospheric INPs is a tiny fraction of total ambient aerosol particles, and there may be a chance that the foreign compound(s), which would not be detectable with FIGAERO-CIMS and coincidently/inherently mixed with lactic acid, trigger(s) ice nucleation. In the reviewer's opinion, this statement is only valid if we know all the ambient droplets have a size smaller than the GCVI cut-off size (<6-7 micron, right?) and if the authors were able to extract only ice residuals. The reviewer presumes this was not the case as the ambient droplet size typically exceeds 10 micron in diameter, and the cut-size of GCVI is not flexible to extract the large particle residuals. Please re-clarify and elaborate on this point.**

The authors thank the reviewer for raising this point. We would first like to point out that we do not discriminate between liquid and solid cloud particles in the GCVI. With the cut-off size of 6-7 µm we discriminate both liquid cloud droplets and ice cloud particles from interstitial aerosols. Also, with our setup we cannot state how large the fractions of cloud droplets and ice crystals in the sampled cloud residuals are. The only data available that provides qualitative information if the cloud was a mixed-phase, liquid or ice cloud is from the cloud radar. We added to L138-139: "With our setup we cannot state how large the fractions of droplets and ice crystals were. The only available instrumentation to differentiate between a mixed-phase, liquid or ice cloud is the cloud radar." As the cloud measured in this cloud case is most likely a mixed-phase cloud (as shown by the cloud radar data in Fig. 7c) the measured composition can refer to cloud droplets and/or ice crystals.

The authors did not find any literature investigating lactic acid as a source for INP, and the reviewer's point of view that lactic acid acting as INP is rather unlikely seems valid. As the authors state in L637-639, lactic acid is also a tracer for human activity, and could simply be an indication that the GCVI line was touched by humans during maintenance. As the chemical composition data from *before* and *after* is sampled via the whole air inlet and *during* is sampled via the GCVI, the elevated signal of lactic acid *during* could therefore just arise from the different inlet systems. The GCVI needed more maintenance than the whole air inlet; hence, also higher levels of lactic acid could arise as a result, which would then be reflected in the chemical composition data of the cloud residuals.

To get a better idea if the lactic acid signal is an actual signal or rather background related to human activity, we investigated the cloud residual blanks during the entire year of the NASCENT campaign including the ambient data, as well as the whole air inlet blanks which were sampled at time points close to the respective cloud residual blanks. We could not identify a clear difference in the thermogram shapes between the cloud residual blank and the whole air blank neighbors. Elevated signals of lactic acid might therefore be actual signal, but we can still not exclude potential handling effects. Other

products that might help to investigate this are palmitic and stearic acid. We discuss their signal together with lactic acid further down in the comment from reviewer 2 on Line 340, and decide to include all three compounds in our analysis, but also present how the chemical composition looks when excluding them in the supplement.

**L593-594 & Fig. 8c during: What is the implication of a virtually equal MSA signal to the SA signal here? Based on the previous statement (L454-456), wouldn't it infer that aerosols measured for the 'during' period is dominantly anthropogenic?**

The authors want to point out that the MSA and SA signals are plotted on different y-axes. The reviewer is right that the statement in L509-511 indicates that the measured aerosol particles *during* are of anthropogenic origin. However, SA can also be formed in the cloud droplets by aqueous-phase oxidation of $SO_2$ (Hoppel et al., 1994), which can be of marine origin as well. The authors changed L654-656 to "This decrease in the cloud residuals is almost entirely driven by changes in SA, and can either indicate that *during* is influenced by anthropogenic sources, or that SA is formed in the cloud droplets through aqueous-phase oxidation of $SO_2$."

**[Technical comments]**

**P3L109: The abbreviation of NASCENT should be given in its first appearance – L103.**

The authors moved the abbreviation to its first appearance in Sect. 2, L111-112.

**L237: --> Karlsson et al. (2021),**

The authors adjusted the citation.

**L576: INP is abbreviated in L43. The authors may double-check all the abbreviations in this manuscript.**

The authors deleted the full name of the abbreviation of INP in L637 and double-checked all abbreviations.

**References**

Gong, X., Wex, H., Voigtländer, J., Fomba, K. W., Weinhold, K., van Pinxteren, M., Henning, S., Müller, T., Herrmann, H., and Stratmann, F.: Characterization of aerosol particles at Cabo Verde close to sea level and at the cloud level – Part 1: Particle number size distribution, cloud condensation nuclei and their origins, Atmos. Chem. Phys., 20, 1431–1449, https://doi.org/10.5194/acp-20-1431-2020, 2020.

Hoppel, W. A., Frick, G. M., Fitzgerald, J. W., and Wattle, B. J.: A Cloud Chamber Study of the Effect That Nonprecipitating Water Clouds Have on the Aerosol Size Distribution, Aerosol Science and Technology, 20, 1–30, https://doi.org/10.1080/02786829408959660, 1994.

Karlsson, L., Krejci, R., Koike, M., Ebell, K., and Zieger, P.: A long-term study of cloud residuals from low-level Arctic clouds, Atmos. Chem. Phys., 21, 8933–8959, https://doi.org/10.5194/acp-21-8933-2021, 2021.

World Meteorological Organization: Guide to meteorological instruments and methods of observation, 7th ed., World Meteorological Organization, Geneva, Switzerland, 2008.

**RC2:**

**The authors investigate cloud residual characteristics based on a one-year-long in-situ measurement in the Arctic, and analyze seasonal variations of the number size distributions and molecular composition of the cloud residual samples. They also used air mass back trajectories to trace back the potential source regions and explain the differences among different cloud residual samples. In addition, they compared the size and composition changes for a cloud case study before, during and after cloud processing to study the cloud impact.**

**The manuscript is well written and in particular the methodology is very well described. Molecular level studies of aerosol particles for Arctic, in particular of organic species and inorganic acids in cloud residuals are very scarce but important to understand CCN properties and aerosol-cloud interactions. I would thus recommend the publication of the manuscript in ACP after minor revisions.**

We thank the reviewer for their time to review our work, and for the recommendation to publish our manuscript. In the following we respond to each of the reviewer's comments (in bold) individually (normal font). Line numbers in the responses refer to the revised document. We also include the modifications of revised figures. The revised figures are numbered as they appear in the manuscript, while figures shown to the reviewer are numbered with R.

**Specific**

**Line 53: How about the autumn season? Is it similar as in summer or the winter/spring?**

We thank the reviewer for pointing out this oversight. We modified L55-60 as following: "Overall, the Arctic aerosol particle number, size and composition follow a distinct annual cycle, where long-range atmospheric transport dominates the accumulation mode particles in winter and spring, and frequent new particle formation in the summer the Aitken mode abundance. Fall is the cleanest season with the lowest particle number and mass concentrations, with only few accumulation mode particles (Tunved et al., 2013). Relatively speaking, the aerosol composition is mainly dominated by sea salt and long range transport from lower latitudes in the winter, while organics of biogenic origin and sulfate are becoming increasingly important in late spring and summer (Moschos et al., 2022a)."

**Line 59: It seems contradictory to the statement in Line 53 about local sources being more important in (brighter) summer.**

The authors thank the reviewer for indicating that there is a contradictory statement in L59 in the first version of the manuscript. The reviewer is right, the statement that "local sources increase in importance during the summer" actually referred to only the organic fraction, and thus excluded sea salt. The authors kept the correct statement in L65-66 ("Sea salt is the largest contributor to particulate matter by mass across the Arctic. Its relative contribution to total particulate matter has been shown to be higher in the dark period compared to the bright season (Moschos et al., 2022a)") and revised L55-60 as explained in the response to the previous comment ("Overall, the Arctic aerosol particle number, size and

composition follow a distinct annual cycle, where long-range atmospheric transport dominates the accumulation mode particles in winter and spring, and frequent new particle formation in the summer the Aitken mode abundance. Fall is the cleanest season with the lowest particle number and mass concentrations, with only few accumulation mode particles (Tunved et al., 2013). Relatively speaking, the aerosol composition is mainly dominated by sea salt and long range transport from lower latitudes in the winter, while organics of biogenic origin and sulfate are becoming increasingly important in late spring and summer (Moschos et al., 2022a))". The authors also made the respective correction for the organics in L72-73 "The organic fraction of Arctic aerosol is dominated by anthropogenic sources in winter, and natural emissions increase in importance in summer."

**Line 68: Where does the trimethylamine come from? Is it also from marine like DMS?**

Trimethylamine can have a variety of sources (see e.g. review by Ge et al., 2011) among which the ocean is suggested by Willis et al. (2016) to provides an important source. We added the information about the marine source of this compound to L73-76 "In summer, the growth of aerosol particles has been associated with the presence of methanesulfonic acid (MSA), produced from the oxidation of dimethylsulfide (DMS) released by marine phytoplankton, marine trimethylamine ($N(CH_3)_3$) and other organic compounds (Willis et al., 2016; Beck et al., 2021).".

**Line 94: Could the authors add the reason why the cloud residuals have this kind of pattern?**

The reason behind the varying composition of the cloud residuals compared to ambient aerosol particles below and above 0 °C could be linked to the INP ability of mineral dust and sea spray. The authors modified L99-102 to "They focused on the fraction of sea salt, mineral dust, sulfate, K-bearing and carbonaceous material-containing particles and found that the cloud residuals have the same composition as ambient aerosol particles at positive temperatures, while at negative temperatures the residuals contained more mineral dust and sea salt compared to the ambient aerosol, likely reflecting the good INP ability of mineral dust and sea spray.".

**Line 291: It seems contradictory to the finding from Karlsson et al. (2022) in Line 283 which found dominating fraction of Aitken mode particles in winter and accumulation mode in spring/summer. Please double check.**

The authors thank the reviewer for pointing out this contradiction. The reviewer is right, what the authors wrote in L291 in the first version of the manuscript should refer to Tunved et al. (2013), who focused on the ambient aerosol number and size distribution. Karlsson et al. (2021) only analyzed number size distributions for the cloud residuals. The authors corrected L324-332 to "Also, the ambient aerosol size distributions largely follow the findings from a previous long-term study on ambient aerosol number size distributions at this site (Tunved et al., 2013) with a dominating accumulation mode in winter and fall, and a shift to more Aitken mode dominated distributions in summer. The number size distributions for the cloud residuals agree with Karlsson et al. (2021) to some extent: they agree insofar that they both show accumulation mode cloud residuals dominating in spring (144 nm on May 18, 2020), and Aitken mode cloud residuals dominating in winter (18 nm on Dec 25, 2020). In summer and late fall our cloud

residuals show a peak in the Aitken mode (56 nm on Jun 12-1, 2020 and plateau from around 66 to 144 nm in late fall), while Karlsson et al., (2021) show that cloud residuals in these seasons are more dominated by accumulation mode particles. However, the cloud residuals in our study also exhibit a peak in the accumulation mode in summer, with lower number concentrations compared to the Aitken mode."

**Line 313: Do you mean particles below 20nm are captured in GCVI as cloud residuals due to wake effect? Please rephrase/clarify a bit more.**

L349-351 was clarified as following: "Other possible reasons could be that in the dry counterflow air hygroscopic particles shrink to sizes much smaller than they have at ambient high humidity conditions, the capture of smaller particles by larger particles due to the wake effect (Pekour and Cziczo, 2011), or entrainment of drier air (Targino et al., 2007)."

**Line 340: Where does the fatty acids come from? Natural sources? They seems quite abundant in winter cloud residuals in Fig 4a, contradictory to the statement in Line 373 that wintertime is dominated by anthropogenic emissions.**

The fatty acids can originate from natural sources, e.g. they can come from the sea surface microlayer and can be ejected into the atmosphere via sea spray emissions. To make this clear already in Sect. 3.3 the authors modified L378-380 to "A subgroup of ICHO and CHO are likely fatty acids (FA), with $h <$ 2, $3 < i < 29$, $k = 2$, $0 < j < 2*i$, representing a natural origin as they can be released into the atmosphere via sea spray emissions (Mashayekhy Rad et al., 2018)."

As pointed out by the reviewer, fatty acids seem to be abundant in the winter cloud residual in Fig. 4a. In the answer to reviewer 1 regarding L575-577 of the first version of the manuscript, we already mentioned that next to lactic acid also palmitic ($C_{16}H_{32}O_2$) and stearic acid ($C_{18}H_{36}O_2$), which are both also fatty acids, can come from handling of the GCVI, as they are in hygiene products. To check whether we established the backgrounds for all three of these compounds in a good way, we looked at all cloud residual samples and blanks from the entire NASCENT year. We clearly see that palmitic and stearic acid are above the background in the Dec 25, 2019 cloud residual. Also, the whole air blanks in the week of Dec 25 show high signals for stearic, palmitic and lactic acid, which suggests that it might be a possibility that lactic, palmitic and stearic acid are just in the background. However, as the scaling of the blanks (Sect. S2) should have taken care of the high background, it cannot be ruled out that these compounds have indeed signal during that part of the year. Palmitic and stearic acid have previously been observed in sea surface microlayer samples from the Arctic Ocean (Mashayekhy Rad et al., 2018); albeit in the summer time, where there was probably a more active sea surface microlayer. How well this can be transferred to our winter December case is unclear.

Overall, it is not clear if the observed signal of palmitic, stearic and lactic acid is real (i.e. from the ambient air) or if heavily influenced by background. To show what the impact of excluding lactic, palmitic and stearic acid is on the overall chemical composition of the cloud residuals during the year, we added the absolute signal of CHO+ICHO and the fatty acids for the winter samples to Fig. 5, and prepared Fig. 5 without these compounds and added it to the supplement (Sect. S6, Fig. S10). We

observe that overall the composition does not change for the majority of our samples when excluding lactic, palmitic and stearic acid. The largest difference occurs for the cloud residual for the two December cases, Dec 25, 2019 and Dec 9, 2020. For Dec 25, 2019 the fatty acid signal is reduced in both the absolute and in the relative signal. For Dec 9, 2020 the CHO+ICHO signal decreases on both an absolute and a relative scale, while the ICHON increase slightly on a relative scale. Additionally, we show the temporal gap between the cloud residual sample and the respective blank used (Sect. S6, Fig. S9). We observe the highest signals in the background subtracted cloud residual signals for those with the largest gap in time between blank and sample collection time, which are the samples in the winter time. In addition, the organic mass concentrations are usually lower in winter compared to summer, so using a blank from a time period with higher mass concentrations (e.g. end of January within the Arctic haze) for times where mass concentrations were lower (e.g. end of December) might add further uncertainty to the final background-subtracted signal. The authors added to Sect. 3.4, L430-433: "We also indicate the absolute and relative signals for the winter samples when excluding the compounds that might be related to handling of the GCVI (lactic, palmitic and stearic acid). For more details about the signals of lactic, palmitic and stearic acid and that excluding these compounds does not change the overall pattern of the chemical composition during the rest of the year see Sect. S6, Fig. S9, S10." And in the supplement the authors added Sect. S6, L110-115: "The chemical formulas referring to lactic acid ($IC_3H_6O_3$-), palmitic acid ($IC_{16}H_{32}O_2$-) and stearic acid ($IC_{18}H_{36}O_2$-) could potentially be related to handling of the GCVI. Based on our data, we were not able to clearly identify if they are only a background signal or if they are actual compounds in the cloud residuals. We observe especially high signals after background subtraction of the compounds in question during the times when the gap in time of sampling and taking a blank was the highest (Dec 25, 2019; Dec 9, 2020, Fig. S9). Fig. S10 shows the chemical composition of the cloud residuals when excluding lactic, palmitic and stearic acid from the absolute and relative signal."

The authors have added a statement about this now also to the manuscript, L394-400: "In contrast to spring, summer and fall, the winter cloud residuals show signal above background for only very few compounds, mainly $IC_3H_6O_3$- (likely lactic acid) and the fatty acids $IC_{16}H_{32}O_2$- (likely palmitic acid) and $IC_{18}H_{36}O_2$- (likely stearic acid). The two latter have been previously observed in the sea surface microlayer from the Arctic Ocean (Mashayekhy Rad et al., 2018). However, these observations were from the summer time high Arctic, and the conditions for winter might not be comparable. In addition, lactic, palmitic and stearic acid might also be attributed to handling of the GCVI during maintenance. We show the impact of these three compounds on the chemical composition of the cloud residuals in Sect. 3.4." Due to two comments further below (about Line 355 in the first version of the manuscript/Fig. 4a) this sentence also has the statement about NA removed.

Accordingly, in Sect. 3.4 we added to L451-453: "The group of fatty acids also includes the previously mentioned palmitic and stearic acid that might be related to hygiene products. Excluding these two compounds would decrease both the absolute and the relative signal of fatty acids in Dec 25, 2020, but the pattern of the rest of the year remains similar (Sect. S6, Fig. S10)."

The authors also thank the reviewer for indicating that the potential natural source of fatty acids observed in the winter cloud residuals is a contradictory statement to the anthropogenic dominated composition in winter compared to summer. The authors want to clarify that this statement is meant to refer to literature and revised L418-421 to "Based on literature, the particle composition in Ny-Ålesund during the dark period is dominated by anthropogenic emissions reaching the station due to atmospheric long-range transport, whereas in the sunlit period of the year, natural emissions can account for almost half of the organic submicron aerosol burden (Moschos et al., 2022a, b)."

[Figure]

Figure 5: (a) Absolute signal of different compound groups (CHO+ICHO, ICHON, $I_{0-1}C_{i=4-28}H_{2i}O_2-$ (fatty acids), $IHNO_3-$ (NA $IC_6H_{10}O_5-$ (levoglucosan), $IH_2SO_4-$ (SA), $ICH_4SO_3-$ (MSA)) in the different cloud residual samples, and the respective $PM_1$ mass. (b) Relative signal of different compound groups in the different cloud residual samples. Note: in the absolute signal view in (a) the CHO+ICHO group contains also the signal of $IC_6H_{10}O_5-$, and $I_{0-1}C_{i=4-28}H_{2i}O_2-$, whereas for the relative signal in (b) the signal from these two groups have been subtracted from CHO+ICHO. The colored horizontal lines in (a) indicate the absolute signals of CHO+ICHO and fatty acids when excluding the compounds that might be related to handling of the GCVI (lactic, palmitic and stearic acid). In analogy, in (b) the colored horizontal lines indicate the relative signal of CHO+ICHO (green), fatty acids (purple) and the ICHON (blue, see Sect. S6, Fig. S10 for the other compound groups) when excluding lactic, palmitic and stearic acid.

[Figure]

Figure S9: (a) Background (BG) subtracted and non-BG subtracted signals of lactic, stearic and palmitic acid and the ratio of their BG subtracted signal to the total organic signal. (b) Background subtracted signals of lactic, stearic and palmitic acid color coded by the time difference between the sample and the blank.

[Figure]

Figure S10: (a) Absolute signal of different compound groups (CHO+ICHO, ICHON, $I_{0-1}C_{i=4-28}H_{2i}O_2$-
(fatty acids), $IHNO_3$- (NA $IC_6H_{10}O_5$- (levoglucosan), $IH_2SO_4$- (SA), $ICH_4SO_3$- (MSA)) in the different
cloud residual samples, and the respective $PM_1$ mass. (b) Relative signal of different compound groups
in the different cloud residual samples. Note: in the absolute signal view in (a) the CHO+ICHO group
contains also the signal of $IC_6H_{10}O_5$-, and $I_{0-1}C_{i=4-28}H_{2i}O_2$-, whereas for the relative signal in (b) the signal
from these two groups have been subtracted from CHO+ICHO. This figure is similar to Fig. 5 from the
main text, but excluding the three compounds that might be linked to hygiene products (lactic, palmitic
and stearic acid).

**Line 343: What is this "IC6H15O7N-" and where is it from? The hydrogen atoms are saturated and seems not to be C6H15O7N clustered with I-.**

The authors do not know what this compound is and where it could come from. The authors observe this compound also as a clear signal in the blanks, which is why it is considered to be a background compound and has been excluded in the analysis. To make this clearer to the reader the authors modified L382-383 to "From the ICHON group we exclude $IC_6H_{15}O_7N-$. It is unclear where this compound is originating from, but since it shows a very high signal in the cloud residual blanks, we attribute it to a background signal.".

**Line 355/Fig. 4a: NA seems to be negative values as a main signal above background in Fig. 4a. Why are signals in mass spectra negative? Due to the blank subtraction?**

The reviewer is right, the signal of NA in the winter cloud residual presented in Fig. 4a is negative (as well as the winter cloud residual given in the supplementary information, Sect. S5, Fig. S8) due to background subtraction. The authors modified the text in L394-396 accordingly: "In contrast to spring, summer and fall, the winter cloud residuals show signal above background for only very few compounds, mainly $IC_3H_6O_3-$ (likely lactic acid) and the fatty acids $IC_{16}H_{32}O_2-$ (likely palmitic acid) and $IC_{18}H_{36}O_2-$ (likely stearic acid)."

To further clarify that the negative signals in the mass spectra occur due to the blank subtraction, the authors added the following statement to L390: "Negative signals occur due to the subtraction of the blank."

**Line 363: This is consistent with the higher RH in summertime in Figure 2b. Could add this info.**

The authors do not understand what the reviewer is suggesting. We do indeed observe high RH levels in the summer when we also observe MSA in the cloud residuals, but the RH shown in Fig. 2 covers the data of the entire month and does not relate to the sampling time of the summer cloud residual presented in Fig. 4c. The authors noticed that the section we are referring to is incorrect and changed L410 to: "In Sect. 3.5 we investigate this further."

**Line 385: What percentage of CHO accounts for CHO+ICHO groups? Do CHO and ICHO have similar trends? Do you also see CHON?**

CHO account for between 2 and 11 percent of the CHO+ICHO group and they do have a similar trend (see Figure R1). We do not see CHON.

[Figure]

Figure R1: Absolute signal of ICHO and CHO for all the cloud residuals.

**Line 393: Similar to what the authors mentioned in Line 430/436 for trajectories for September 12, the higher CHO+ICHO, levoglucosan and NA are mainly of anthropogenic origin. Could add this trajectory info here too.**

We thank the reviewer for pointing this out. The authors added the following information to L441-443: "The elevated levels of CHO+ICHO, NA and levoglucosan in the late summer (Sep 12) indicate an anthropogenic influence, supported by the back trajectories originating from the large anthropogenic source region of sulfur dioxide ($SO_2$) in Kola Peninsula (Sect. S7, Fig. S11)."

**Line 411: levoglucosan doesn't seem to have elevated absolute signals for cloud residuals on May 21 or Dec 9 samples from Fig 5a. Please double check.**

The reviewer is right. We revised the text in L461-462 accordingly to "We observe elevated absolute signals of levoglucosan in the cloud residuals from mid-June until September 12 and lower absolute signals in May and October."

**Line 423, Line 474 and Line 632: With information of higher MSA levels in summer only, it's insufficient to "*confirm*" its role in particle growth, unless growth rate calculations or some correlations are done such as Aitken mode particle conc. vs MSA level. Otherwise, I would suggest to change to e.g. "*indicate*".**

L423: The authors agree with the reviewer's suggestion and changed L423 of the original version of the manuscript, L473-475 in the revised version to "This indicates that DMS oxidation products are relevant to grow aerosol particles to CCN-active sizes, which is what previous ambient aerosol observations at the same measurement location already suggested (Park et al., 2021).".

L474: The authors changed L473-475 of the original version of the manuscript, L529-531 in the revised version to "The combination of the observed dominating Aitken mode and elevated levels of MSA in

the cloud residual samples at the end of June 2020 further indicates that during this time of the year, MSA clearly contributes to the growth of newly formed particles into CCN sizes.".

L623: Here we already use the word "indicate", therefore the text was kept as is.

**Line 511: Is the advection of a different air mass visible also from trajectories, or have some indication from e.g. vertical wind speed measurement? At least it needs a reference to support.**

The authors thank the reviewer for this valid comment. The authors added the following information to L568-572: "The back trajectories indicate that *during,* the air spent less time over the Russian coast compared to *before*, and more time over the Kara Sea (Fig. 7a). Additionally, while most of the time the air mass was above the boundary layer *before, during* and *after*, there are indications in the back trajectories that *during* the air was within the boundary layer when it was passing over the Kara Sea (Sec. S8, Fig. S12a,b). In this region, the air mass could have collected the particles visible as a peak in the number size distribution around 79 nm."

**Line 596: Could the authors comment on the influence of different air mass on the chemical composition of this cloud event, in order for more valid comparison for this cloud event?**

The authors are unsure what is meant by the reviewer, since for this particular cloud event we discuss the back trajectories in detail (L550-553). We specifically chose this event for such a comparison because of the little change in back trajectories throughout the event.

**Technical**

**Line 46: Change to "warming the Arctic".**

This was not changed as suggested because we respectfully believe our version to be the correct one.

**Line 157: Add "inch" after "1/4".**

Changed as suggested.

**Line 312: Change to "Other possible reasons".**

Changed as suggested.

**Line 316: Change to "in the following below".**

Changed to "following below".

**Line 330: Section 3.3 is missing. Please double check the section numbers.**

The authors thank the reviewer for indicating this. The authors changed the numbering of the sections from 3.4 until 3.7.2 from the first version of the manuscript to 3.3 until 3.6.2 in the revised version.

**Line 366/Figure 4: The legend colour of different compound families is difficult to differentiate.**

The authors adjusted the color scheme for Fig. 4 and Fig. S8.

[Figure]

Figure 4: Mass spectra of cloud residuals. (a) December 25, 2019, (b) May 18, 2020, (c) June 12-1, 2020, (d) October 28, 2020. The detected compounds are presented as I-clusters, as detected by the FIGAERO-CIMS.

[Figure]

Figure S8: Mass spectra of all cloud residual samples not shown in the main text.

**Line 397: Change "Compounds in the ICHON group might have been formed via ICHO and NOx emissions" to "CHON compounds might be related to oxidations of CH(O) and NOx emissions".**

Changed as suggested.

**Line 464: Change to "a high number contribution" to avoid misunderstandings.**

Changed as suggested.

**Line 549: Delete "to" from "during to the cloud event".**

Changed as suggested.

**Line 565: Change "chemical formulae of" to "chemical formulae corresponding to".**

Changed as suggested.

**References**

Beck, L. J., Sarnela, N., Junninen, H., Hoppe, C. J. M., Garmash, O., Bianchi, F., Riva, M., Rose, C., Peräkylä, O., Wimmer, D., Kausiala, O., Jokinen, T., Ahonen, L., Mikkilä, J., Hakala, J., He, X., Kontkanen, J., Wolf, K. K. E., Cappelletti, D., Mazzola, M., Traversi, R., Petroselli, C., Viola, A. P., Vitale, V., Lange, R., Massling, A., Nøjgaard, J. K., Krejci, R., Karlsson, L., Zieger, P., Jang, S., Lee, K., Vakkari, V., Lampilahti, J., Thakur, R. C., Leino, K., Kangasluoma, J., Duplissy, E., Siivola, E., Marbouti, M., Tham, Y. J., Saiz-Lopez, A., Petäjä, T., Ehn, M., Worsnop, D. R., Skov, H., Kulmala, M., Kerminen, V., and Sipilä, M.: Differing Mechanisms of New Particle Formation at Two Arctic Sites, Geophys Res Lett, 48, https://doi.org/10.1029/2020GL091334, 2021.

Ge, X., Wexler, A. S., and Clegg, S. L.: Atmospheric amines – Part I. A review, Atmospheric Environment, 45, 524–546, https://doi.org/10.1016/j.atmosenv.2010.10.012, 2011.

Karlsson, L., Krejci, R., Koike, M., Ebell, K., and Zieger, P.: A long-term study of cloud residuals from low-level Arctic clouds, Atmos. Chem. Phys., 21, 8933–8959, https://doi.org/10.5194/acp-21-8933-2021, 2021.

Mashayekhy Rad, F., Leck, C., Ilag, L. L., and Nilsson, U.: Investigation of ultrahigh-performance liquid chromatography/travelling-wave ion mobility/time-of-flight mass spectrometry for fast profiling of fatty acids in the high Arctic sea surface microlayer, Rapid Commun Mass Spectrom, 32, 942–950, https://doi.org/10.1002/rcm.8109, 2018.

Moschos, V., Schmale, J., Aas, W., Becagli, S., Calzolai, G., Eleftheriadis, K., Moffett, C. E., Schnelle-Kreis, J., Severi, M., Sharma, S., Skov, H., Vestenius, M., Zhang, W., Hakola, H., Hellén, H., Huang, L., Jaffrezo, J.-L., Massling, A., Nøjgaard, J. K., Petäjä, T., Popovicheva, O., Sheesley, R.

J., Traversi, R., Yttri, K. E., Prévôt, A. S. H., Baltensperger, U., and El Haddad, I.: Elucidating the present-day chemical composition, seasonality and source regions of climate-relevant aerosols across the Arctic land surface, Environ. Res. Lett., 17, 034032, https://doi.org/10.1088/1748-9326/ac444b, 2022a.

Moschos, V., Dzepina, K., Bhattu, D., Lamkaddam, H., Casotto, R., Daellenbach, K. R., Canonaco, F., Rai, P., Aas, W., Becagli, S., Calzolai, G., Eleftheriadis, K., Moffett, C. E., Schnelle-Kreis, J., Severi, M., Sharma, S., Skov, H., Vestenius, M., Zhang, W., Hakola, H., Hellén, H., Huang, L., Jaffrezo, J.-L., Massling, A., Nøjgaard, J. K., Petäjä, T., Popovicheva, O., Sheesley, R. J., Traversi, R., Yttri, K. E., Schmale, J., Prévôt, A. S. H., Baltensperger, U., and El Haddad, I.: Equal abundance of summertime natural and wintertime anthropogenic Arctic organic aerosols, Nat. Geosci., https://doi.org/10.1038/s41561-021-00891-1, 2022b.

Park, K., Yoon, Y. J., Lee, K., Tunved, P., Krejci, R., Ström, J., Jang, E., Kang, H. J., Jang, S., Park, J., Lee, B. Y., Traversi, R., Becagli, S., and Hermansen, O.: Dimethyl Sulfide-Induced Increase in Cloud Condensation Nuclei in the Arctic Atmosphere, Global Biogeochem Cycles, 35, https://doi.org/10.1029/2021GB006969, 2021.

Pekour, M. S. and Cziczo, D. J.: Wake Capture, Particle Breakup, and Other Artifacts Associated with Counterflow Virtual Impaction, Aerosol Science and Technology, 45, 758–764, https://doi.org/10.1080/02786826.2011.558942, 2011.

Targino, A. C., Noone, K. J., Drewnick, F., Schneider, J., Krejci, R., Olivares, G., Hings, S., and Borrmann, S.: Microphysical and chemical characteristics of cloud droplet residuals and interstitial particles in continental stratocumulus clouds, Atmospheric Research, 86, 225–240, https://doi.org/10.1016/j.atmosres.2007.05.001, 2007.

Tunved, P., Ström, J., and Krejci, R.: Arctic aerosol life cycle: linking aerosol size distributions observed between 2000 and 2010 with air mass transport and precipitation at Zeppelin station, Ny-Ålesund, Svalbard, Atmos. Chem. Phys., 13, 3643–3660, https://doi.org/10.5194/acp-13-3643-2013, 2013.

Willis, M. D., Burkart, J., Thomas, J. L., Köllner, F., Schneider, J., Bozem, H., Hoor, P. M., Aliabadi, A. A., Schulz, H., Herber, A. B., Leaitch, W. R., and Abbatt, J. P. D.: Growth of nucleation mode particles in the summertime Arctic: a case study, Atmos. Chem. Phys., 16, 7663–7679, https://doi.org/10.5194/acp-16-7663-2016, 2016.

**Other things modified that are not pointed out by the reviewers:**

In the original version L361-361, now L406-407, were changed to "The cloud residuals observed in spring, summer and early fall (May 18, 2020 until Sep 12, 2020) show a clear MSA signal, whereas the cloud residuals observed in late fall and winter (Dec 25, 2019; Oct 28 and Dec 9, 2020) do not (Sect. S5, Fig. S8)."

Appearance of Table 1.

Fig. 4: Updated the molecular formulas to the format given in the main text (ICHOX instead of CHIOX), enlarged the legend and adjusted the y-axis.

Fig. 8c: Updated y-labels label for the two y-axes on the right, see below.

[Figure]

Figure 8: Cloud case on May 18, 2020, chemical composition of total particles before, and after, of cloud residuals during the cloud event. (a) Pie chart showing the relative contribution of groups and individual compounds to the measured FIGAERO-CIMS signal. Green: CHO+ICHO. Purple: fatty acids, blue: ICHON, yellow: $ICH_4O_3S$- (MSA), red: $IH_2O_4S$- (SA), grey: $IHNO_3$- (NA). Bar chart with absolute signal of CHO+ICHO compounds before, during and after the cloud event grouped by the number of carbon (1-19) and oxygen (O1-O10) atoms. (b) Mass defect highlighting individual compounds that show an increasing or decreasing trend during the cloud event (all: all compounds detected with the FIGAERO-CIMS; increasing: compounds of which the relative signal is increasing from before, via during until after the cloud; decreasing: compounds of which the relative signal is decreasing from before, via during until after the cloud; highest during: compounds of which the relative signal is highest during the cloud compared to before and after). (c) Ratio of MSA to sulfuric acid (SA) and their normalized signal to the total signal before, during and after the cloud event.

Supplement figure trajectories: label of Jun 14-1 changed to Jun 14.

Change of the email address and the affiliation of the corresponding author.

Supplement Table S1: The date of the blank from September 13 used was added.